# Diode effect in Josephson junctions with a single magnetic atom

Martina Trahms[1], Larissa Melischek[2], Jacob F. Steiner[2], Bharti Mahendru[1], Idan Tamir[1], Nils Bogdanoff[1], Olof Peters[1], Gaël Reecht[1], Clemens B. Winkelmann[3], Felix von Oppen[2] & Katharina J. Franke[1✉]

Current flow in electronic devices can be asymmetric with bias direction, a phenomenon underlying the utility of diodes[1] and known as non-reciprocal charge transport[2]. The promise of dissipationless electronics has recently stimulated the quest for superconducting diodes, and non-reciprocal superconducting devices have been realized in various non-centrosymmetric systems[3–10]. Here we investigate the ultimate limits of miniaturization by creating atomic-scale Pb–Pb Josephson junctions in a scanning tunnelling microscope. Pristine junctions stabilized by a single Pb atom exhibit hysteretic behaviour, confirming the high quality of the junctions, but no asymmetry between the bias directions. Non-reciprocal supercurrents emerge when inserting a single magnetic atom into the junction, with the preferred direction depending on the atomic species. Aided by theoretical modelling, we trace the non-reciprocity to quasiparticle currents flowing by means of electron–hole asymmetric Yu–Shiba–Rusinov states inside the superconducting energy gap and identify a new mechanism for diode behaviour in Josephson junctions. Our results open new avenues for creating atomic-scale Josephson diodes and tuning their properties through single-atom manipulation.

Since the invention of semiconductor p–n junctions, currents asymmetric in the direction of the applied bias voltage have been central to the development of electronic devices[1]. In p–n junctions, non-reciprocal charge transport emerges from the band misalignment at the interface, which breaks inversion symmetry. In the absence of abrupt material interfaces, non-reciprocal charge transport usually occurs when broken inversion symmetry (for example, by an electric field or the Rashba effect) is accompanied by broken time-reversal symmetry (for example, by an applied magnetic field)[2]. If the current flows perpendicular to crossed electric and magnetic fields, its magnitude depends on the direction, a phenomenon known as the magnetochiral effect[11].

Non-reciprocal charge transport is particularly appealing for superconducting devices. They can exhibit dissipationless supercurrent in one direction, whereas the reverse direction is resistive, allowing for essentially unlimited resistance ratios. Diode behaviour has recently been realized in non-centrosymmetric low-dimensional superconductors[3,4,9], as well as in inversion-symmetry-breaking stacks of different superconductors[5], making use of the strong magnetochiral effect when spin–orbit coupling and superconducting gap are of comparable magnitude. The need for a time-reversal-breaking external magnetic field can be avoided by including magnetic interlayers[12].

Josephson junctions provide an alternative platform for diode-like behaviour in superconductors, offering further tunability and potentially interfacing with superconducting qubits. Although two or more Josephson junctions combined into superconducting quantum interference devices (also known as SQUIDS) have long been proposed as

amplifiers and rectifiers[13,14], experiments on single Josephson junctions have only recently observed non-reciprocal behaviour. Baumgartner et al.[6] used a proximity-coupled two-dimensional electron gas with strong spin–orbit interaction, Pal et al.[7] observed diode-like behaviour in superconducting junctions in proximity to a topological semimetal and Diez-Merida et al.[8] in twisted bilayer graphene. Although these devices required external magnetic fields to induce the diode effect, Wu et al.[10] demonstrated rectification in a NbSe$_2$/Nb$_3$Br$_8$/NbSe$_2$ junction without magnetic fields[15].

Here we report that insertion of a single atom can induce diode-like behaviour in Josephson junctions implemented using a scanning tunnelling microscope (STM). Josephson coupling with and without adatoms has long been investigated using STMs with superconducting tips, focusing on spectroscopy of tunnelling processes and excitations[16–18], pair-density waves[19], phase diffusion[20], photon-assisted tunnelling[21–23], Josephson spectroscopy[24,25] and 0–π transitions[26]. Although previous work on single-atom junctions focused on voltage-biased junctions, diode effects require current-biased measurements. We realize current-biased Josephson junctions and find diode-like behaviour when including a single magnetic atom. We show that magnitude and sign of the diode effect can be tuned by the choice of atomic species. This makes our single-atom Josephson diodes a promising platform for studies of superconducting diodes, in particular when combined with single-atom manipulation to assemble the atoms into nanostructures.

We also demonstrate that the non-reciprocity of our atomic-scale Josephson junctions is because of a new mechanism. The current-biased

[1]Fachbereich Physik, Freie Universität Berlin, Berlin, Germany. [2]Dahlem Center for Complex Quantum Systems, Fachbereich Physik, Freie Universität Berlin, Berlin, Germany. [3]Université Grenoble Alpes, CNRS, Institut Néel, Grenoble, France. ✉e-mail: franke@physik.fu-berlin.de

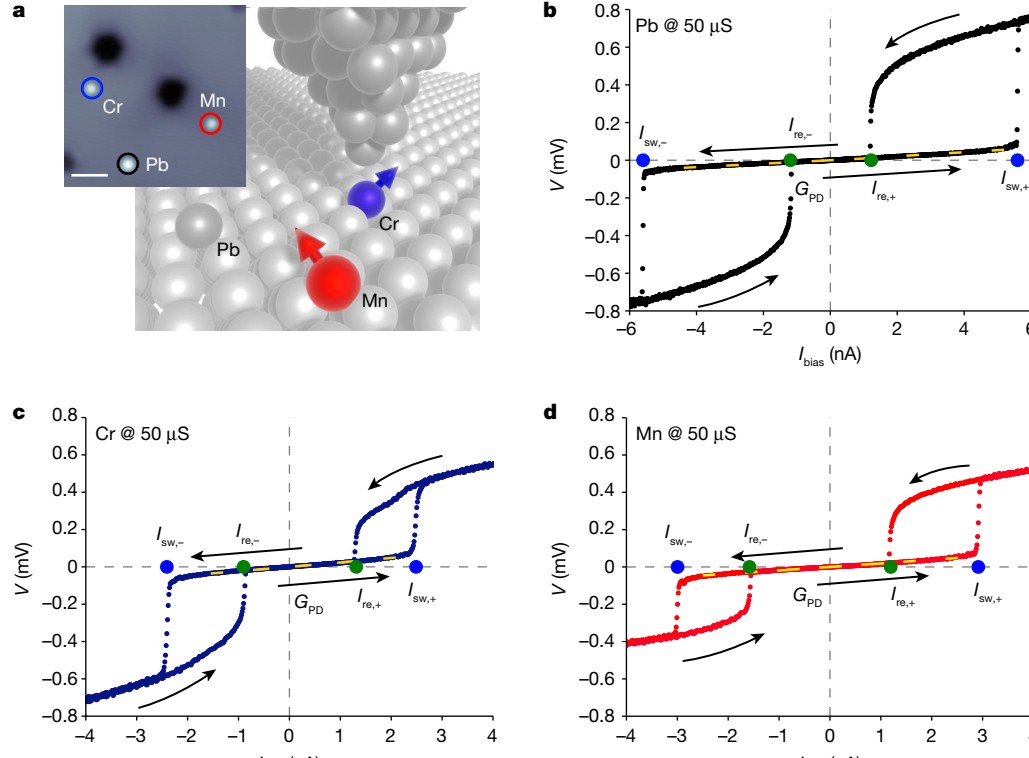

**Fig. 1 | Single-atom Josephson junctions including Pb, Mn and Cr atoms.**
**a**, Sketch of STM-based Josephson junction including a single atom. Inset, STM topography of a Pb(111) surface with individual Pb, Mn and Cr adatoms (coloured circles); scanning parameters: 50 mV, 50 pA. Scale bar, 3 nm.
**b**–**d**, $V$–$I$ curves of current-biased Pb–Pb junctions including a Pb (**b**), Cr (**c**) and

Mn (**d**) atom, measured at a normal-state conductance of $G_N = 50$ μS. Sweep directions are indicated by black arrows and switching and retrapping events by blue and green dots, respectively. The slope at small currents (inverse of the phase-diffusion conductance $G_{PD}$) is marked by a yellow dashed line.

junctions exhibit a hysteretic voltage response[27–30], with the switching current ($I_{sw}$)—marking the transition from dissipationless to resistive junction behaviour on increasing the current bias—well separated from the retrapping current ($I_{re}$)—marking the reverse transition on reducing the current. We find a dominant non-reciprocity in the retrapping current, whereas all previous experiments on hysteretic Josephson junctions found a dominant non-reciprocity in the switching current. We explain this by a new mechanism for non-reciprocity, which is the result of asymmetric quasiparticle damping and does not require breaking of time-reversal symmetry by an applied magnetic field. This is in contrast to strong asymmetries in the switching current, which result from asymmetric current–phase relations.

## Current-biased single-atom Josephson junctions

Figure 1a shows a sketch of our experimental setup. The Josephson junction is formed between the superconducting Pb tip of a STM and an atomically clean superconducting Pb(111) crystal with single Pb, Cr and Mn atoms deposited on its surface (see STM image in Fig. 1a). Advancing towards these atoms by the tip allows us to investigate the influence of individual atoms on otherwise identical Josephson junctions. To establish these atomic-scale Josephson junctions, we advance the STM tip to the surface at a bias voltage well outside the superconducting gap, until a normal-state junction conductance of 50 μS, on the order of but smaller than the conductance quantum, is reached. We then introduce a large resistor (1 MΩ) in series with the junction, such that we effectively control the current bias of the junction.

We first focus on junctions stabilized by a single Pb atom (Fig. 1b). When reducing the bias current from large positive currents, we observe a sharp reduction in the voltage drop across the junction at

the retrapping current ($I_{re} \approx 1.2$ nA). This marks the transition from resistive behaviour dominated by quasiparticle tunnelling (dissipative branch) to the near-dissipationless low-voltage state dominated by Cooper-pair tunnelling. Further reducing and eventually reversing the current bias to negative values, the junction abruptly transitions back to the dissipative branch at the switching current ($I_{sw} \approx -5.6$ nA). Inverting the sweep direction of the current, the $V$–$I$ behaviour exhibits a substantial hysteresis, but for pristine Pb–Pb junctions, the magnitudes of the switching and retrapping currents are independent of the bias direction (Fig. 1b).

## Non-reciprocal Josephson currents

The Josephson junctions exhibit qualitatively different behaviour when the Pb atom is replaced by a single Cr or Mn atom (Fig. 1c,d). Incorporating a single magnetic atom into the junction substantially reduces the switching current compared with the pristine Pb junctions. This is consistent with a reduction of the Josephson peak in voltage-biased measurements on magnetic atoms[24–26]. Notably, we observe that the retrapping current and, to a much lesser extent, the switching current now depend on the direction of the current bias, so that the incorporation of a single magnetic atom makes the junction non-reciprocal. The behaviour of our atomic-scale junctions differs qualitatively from observations of non-reciprocity in larger-scale junctions. Although we observe the dominant asymmetry in the retrapping current, refs. [7,10] find stronger non-reciprocal behaviour in the switching currents.

Next, we directly compare the switching and retrapping currents for both bias directions over a range of junction conductances (Fig. 2a,b). Accounting for the statistical nature of the switching and retrapping processes, every data point averages the switching or retrapping

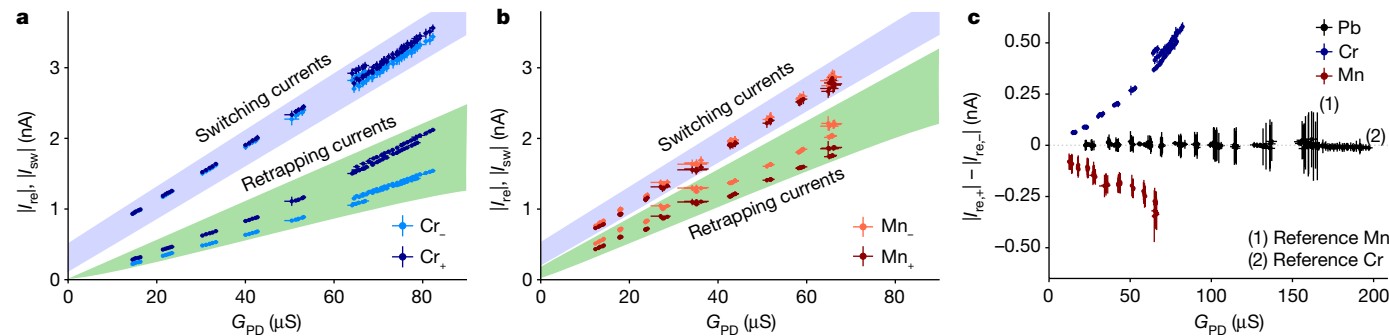

**Fig. 2 | Non-reciprocity of switching and retrapping currents versus junction transparency. a,b**, Absolute values of retrapping and switching currents as extracted from $V$–$I$ curves for Cr (**a**) and Mn (**b**) junctions. Each data point averages over 100 sweeps recorded during longer measurement series started at normal-state conductances between 25 μS and 50 μS (**a**) and 20 μS and 50 μS (**b**) at 10 mV. Error bars indicate the standard deviation of the average values. Although piezoelectric creep slowly changes $G_{PD}$ (determined for each individual sweep), $G_{PD}$ remains essentially constant for the sweeps entering

into a single data point (see Methods for details). Positive/negative current bias is indicated by dark/bright colours and labelled as $Cr_+$/$Cr_-$ and $Mn_+$/$Mn_-$. Panels include data from several measurement times with different samples and tips to highlight the robustness of the effect, apart from small variations in the noise characteristics. **c**, Asymmetry $\Delta I_{re} = |I_{re,+}| - |I_{re,-}|$ of the retrapping current for single-atom Cr, Mn and Pb junctions. Pb junctions exhibit symmetric retrapping currents, whereas Cr and Mn atoms show non-reciprocities of opposite sign.

current over 100 current sweeps (see Extended Data Figs. 2–4 for full histograms). We quantify the junction conductance by the (inverse) slope of the $V$–$I$ curves in the low-voltage regime (compare Fig. 1b–d), which we refer to as the phase-diffusion conductance $G_{PD}$ for reasons explained below. We find that the retrapping currents depend not only on the bias direction but also on the particular type of magnetic atom. For Cr atoms, the retrapping current is much larger in magnitude at positive bias ($I_{re,+}$) than at negative bias ($I_{re,-}$). For Mn atoms, the situation is just reversed. This is further illustrated in Fig. 2c, which shows the asymmetry $\Delta I_{re} = |I_{re,+}| - |I_{re,-}|$ in the retrapping current as a function of $G_{PD}$. Furthermore, there is a considerably weaker asymmetry of the switching current (see also histograms in Extended Data Fig. 1).

## Phase dynamics

The hysteretic junction dynamics can be described within the model of a resistively and capacitively shunted Josephson junction (RCSJ). In this model[27–30], the bias current $I_{bias}$ applied to the junction splits between a capacitive current $I_c = C\dot{V}$, a dissipative current $I_d$ and its associated Nyquist noise $\delta I$, as well as the supercurrent $I_s(\varphi)$. Using the Josephson relation $V = \hbar\dot{\varphi}/2e$ for the voltage $V$ across the junction and assuming ohmic dissipation, $I_d = V/R$, the superconducting phase difference $\varphi$ across the junction can be described as a Brownian particle moving in a tilted washboard potential (Fig. 3e),

$$(\hbar C/2e)\ddot{\varphi} + (\hbar/2eR)\dot{\varphi} + I_s(\varphi) + \delta I = I_{bias}. \quad (1)$$

The tilted washboard potential subjects the Brownian particle to a constant force associated with the bias current $I_{bias}$, as well as to a periodic force originating from the current–phase relation $I_s(\varphi)$ of the junction. We note that, at our measurement temperature of 1.3 K, the junctions are adequately modelled by classical phase dynamics.

Focusing first on the pristine Pb junctions, the hysteretic behaviour emerges as follows. At small bias currents, the phase is trapped in a minimum of the tilted washboard potential, corresponding to supercurrent flow. Increasing the bias current tilts the washboard potential and lowers the potential barrier for activation of the phase particle out of the minimum. Once the phase particle escapes, it crosses over to a running solution associated with a voltage drop across the junction (switching current). Conversely, when reducing the bias current, inertia makes the phase particle retrap into a minimum only at a smaller current, at which friction balances the energy gained owing to the tilt of the washboard potential (retrapping current). In our junctions, switching occurs long

before the bias current reaches the critical current $I_c$ (estimated at 107 nA based on the Ambegaokar–Baratoff formula[31]), at which the tilted washboard potential loses its minima, indicating the importance of the Nyquist noise $\delta I$. We note that we observe a small voltage drop also in the nominally trapped state at small bias current. This behaviour is familiar for small junctions and a well-understood consequence of frequency-dependent damping[30], leading to residual phase diffusion and the zero-bias conductance $G_{PD}$ (see also Methods).

Although this basic RCSJ model predicts reciprocal dynamics, several extensions are known to support non-reciprocal behaviour. Diode-like behaviour can originate with an asymmetric current–phase relation[6,7,32–36] or nonlinear corrections to the capacitive term associated with the quantum capacitance[37]. An asymmetric current–phase relation implies a non-reciprocal switching current, inconsistent with our observations. Nonlinear corrections to the capacitive term induce asymmetric retrapping currents. However, this requires a junction with strongly asymmetric carrier densities on its two sides, a feature that is absent for our Pb–Pb junctions.

## Origin of non-reciprocity

Non-reciprocity of the retrapping current, coexisting with less asymmetric switching currents, suggests instead that the non-reciprocity originates with the damping properties of the junction. Microscopically, the dissipative current $I_d$ accounts for the quasiparticle current flowing in parallel to the supercurrent, as well as dissipation into the electromagnetic environment. Although the latter is expected to be independent of the bias direction, the quasiparticle current can be non-reciprocal.

The asymmetry of the quasiparticle current is directly accessible in voltage-biased measurements, with a superconducting tip, of the same junctions. Figure 3a,b presents tunnelling spectra on Cr and Mn atoms at small junction conductance (0.125 μS), showing strong subgap resonances of the differential conductance $dI/dV$ (and thus current). As well as the coherence peaks at $(2.72 \pm 0.05)$ mV, we resolve three pairs of conductance peaks, labelled by $(\alpha, \beta, \gamma)$, which we identify with Yu–Shiba–Rusinov (YSR) states[38] within the superconducting energy gap $\Delta$. Peaks occurring at voltages $e|V| < \Delta$ originate from the same states, albeit examined by thermally excited quasiparticles[39]. Although the YSR resonances must occur symmetrically in energy, they need not have symmetric intensities[38,40,41]. We observe that this asymmetry is particularly pronounced for the deepest ($\alpha$) YSR state of Mn. By comparison, Cr exhibits weaker but still well-resolved asymmetries

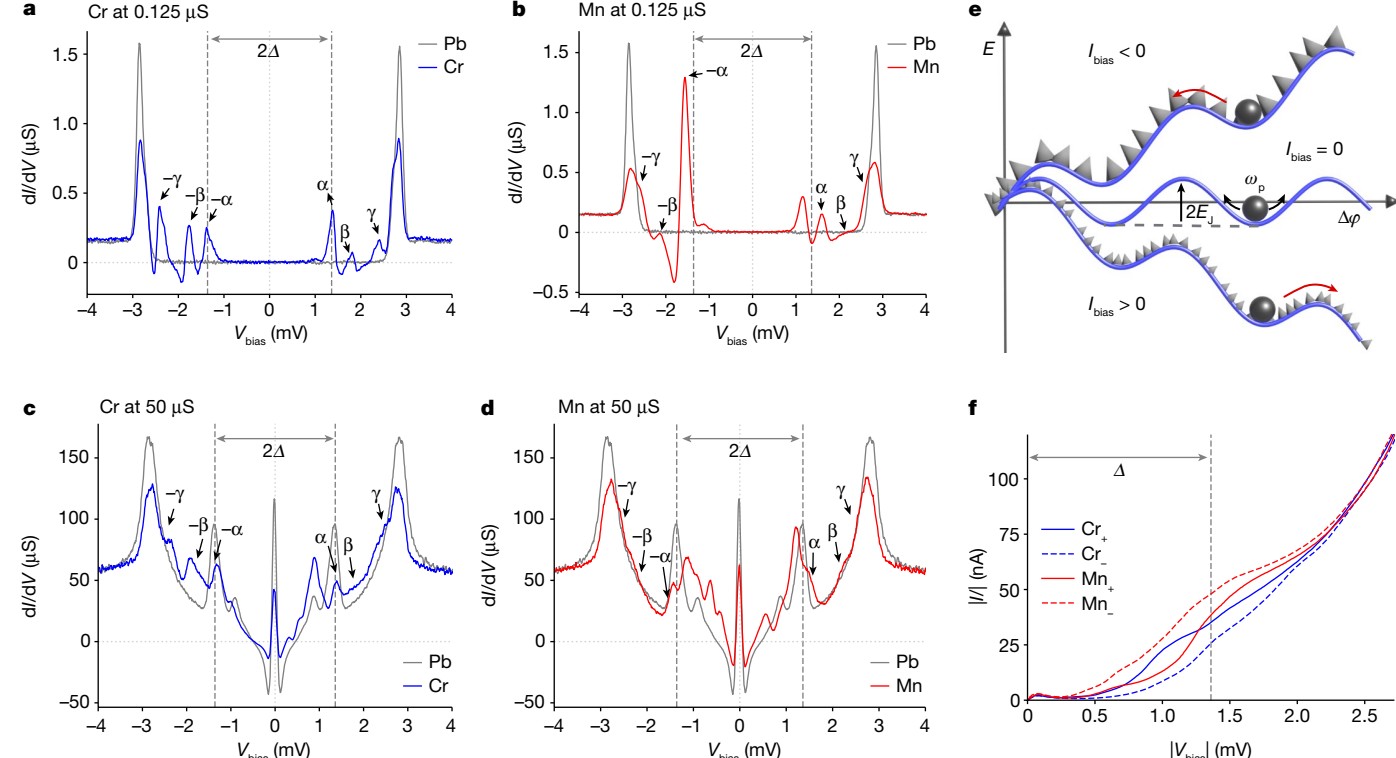

**Fig. 3 | YSR states as origin of non-reciprocity. a,b**, Voltage-biased differential conductance spectra of Cr (**a**) and Mn (**b**) at a normal-state conductance of $G_N = 0.125\ \mu S$. (Conductance set at 500 pA, 4 mV, lock-in modulation $V_{rms} = 15\ \mu V$). The superconducting energy gap of the tip ($\Delta$) is marked by dashed lines. Reference spectra of Pb are shown in grey. YSR states are labelled as α, β and γ. The YSR states are symmetric in energy about zero bias but asymmetric in intensity owing to electron–hole asymmetry. **c,d**, d$I$/d$V$ spectra of the same atoms as in **a** and **b** measured at $G_N = 50\ \mu S$. (Conductance set at 500 nA, 10 mV, lock-in modulation $V_{rms} = 15\ \mu V$). These spectra show a zero-bias Josephson peak as well as several Andreev reflections with and without exciting YSR states. Owing to the electron–hole asymmetry of the YSR states, the spectra exhibit intensities that are distinctly asymmetric about zero bias. **e**, Sketch of

washboard potential (blue line) and friction (roughness of grey background) controlling the dynamics of a current-biased Josephson junction as represented by a phase particle (black spheres). The phase particle can be trapped in a minimum characterized by Josephson energy $E_J$ and plasma frequency $\omega_p$ (trapped state) or slide down the washboard potential (running state). Non-reciprocal behaviour originates with friction, which depends on bias direction, as indicated by the different grey textures. **f**, Current–voltage characteristics of voltage-biased Mn and Cr Josephson junctions for positive (+)/negative (−) voltages at $G_N = 50\ \mu S$. The Cr junctions show a larger current magnitude at positive than at negative bias. The situation is opposite for Mn junctions.

of the YSR-state intensities. Notably, there is no asymmetry in the corresponding d$I$/d$V$ traces for the junction stabilized on a Pb atom (see grey traces in Fig. 3a,b).

These results indicate that the asymmetric subgap conductance associated with the YSR resonances is a natural source of the observed non-reciprocal behaviour. However, the spectra in Fig. 3a,b were taken in the weak-tunnelling regime, in which the YSR resonances are well resolved and are thus not of immediate relevance to the Josephson-junction regime at stronger tunnelling. Figure 3c,d shows d$I$/d$V$ spectra at junction conductances corresponding to the Josephson-junction regime. For the pristine Pb junctions, the larger junction conductance enables further transport processes inside the gap owing to Cooper-pair tunnelling at zero bias (Josephson peak) and several Andreev reflections above the threshold voltages of $eV = 2\Delta/n$ with $n = 2, 3...$ (Fig. 3c,d, grey traces). Consistent with the weak-tunnelling case, the d$I$/d$V$ traces of pristine Pb junctions remain independent of bias direction at high junction conductance.

For the Cr and Mn junctions at higher junction conductance, we observe an even richer in-gap structure, with intensities that are clearly asymmetric in the bias directions. We attribute the extra features to several Andreev processes exciting a YSR state of energy $\varepsilon$ as well as to quasiparticles in the electrodes. These processes have threshold energies of $eV = (\Delta + \varepsilon)/n$ (refs. [24,42]) and reflect the asymmetry of the

underlying YSR states. The resulting asymmetry in the subgap current is shown in Fig. 3f. Notably, the quasiparticle current for Cr is larger at positive bias voltages. Because a larger quasiparticle current implies stronger dissipation, this is consistent with the larger retrapping current for this bias direction of the current-biased Josephson junction. The situation is just reversed for Mn, again consistent with the asymmetry of the retrapping current.

To further corroborate that an asymmetric quasiparticle current can induce non-reciprocal behaviour of Josephson junctions, we perform numerical simulations for an extended RCSJ model[30]. We include frequency-dependent damping, allow for a nonlinear and asymmetric dissipative $I_d(V)$ and account for the Johnson–Nyquist noise associated with the damping. To isolate the effect of asymmetric damping, we extract $I_d(V)$ based on the experimental data in Fig. 3f for the Pb, Cr and Mn junctions but otherwise use identical model parameters (for details, see Methods). Figure 4 shows representative $V–I$ traces, which are symmetric for Pb but exhibit asymmetric retrapping currents for Cr and Mn. The asymmetries clearly reproduce the sign found in the experiment (compare Fig. 1b–d). Consistent with the experiment, our simulations also reproduce a weak asymmetry in the switching current. (The asymmetry of the switching current in Fig. 4 is dominated by statistical fluctuations. The full switching-current histograms shown in Methods have only a much weaker asymmetry.) Finally, we

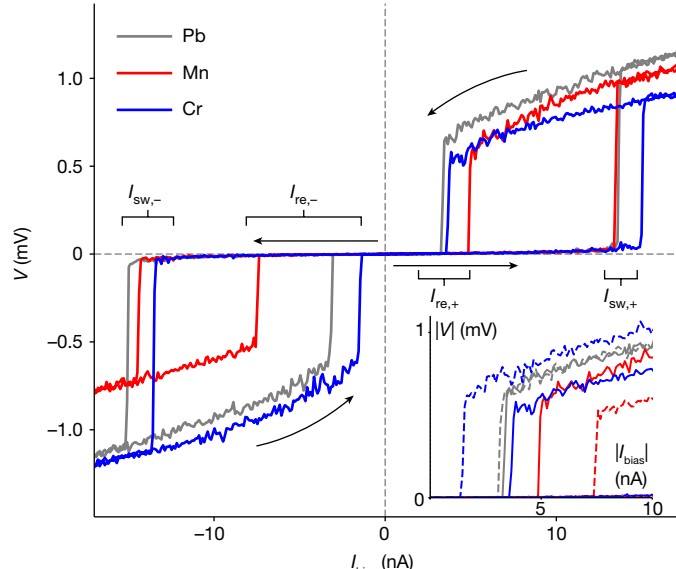

**Fig. 4 | Modelling of non-reciprocity within the RCSJ model.** Simulated hysteretic *V–I* traces based on the extended RCSJ model, accounting for asymmetric as well as frequency-dependent friction. Traces for Pb (grey), Cr (blue) and Mn (red) use the non-ohmic quasiparticle current $I_d(V)$ as extracted from the corresponding experimental data in Fig. 3c,d as input. All other model parameters (see Methods) are identical to highlight the effect of asymmetric friction. Inset, $|V|-|I|$ traces over a smaller range of bias currents using the same data as in the main panel to bring out the asymmetry, showing only the retrapping currents. Solid (dashed) lines correspond to positive (negative) current bias, with colour coding as in the main panel.

comment on the symmetry conditions for non-reciprocity originating from quasiparticle damping. Inversion symmetry is explicitly broken by single-atom junctions with the adatom attached to one of the electrodes. At the same time, the junction is time-reversal symmetric because, in the absence of an external magnetic field, the spin of the magnetic molecule remains unpolarized. Instead, the asymmetric weights of the YSR resonances and hence the non-reciprocity are a consequence of broken particle–hole symmetry (see also Methods).

## Conclusions

Developing device applications for Josephson diodes requires a thorough understanding of the mechanisms underlying their non-reciprocity. By examining the limits of miniaturization, we have created and investigated Josephson diodes whose asymmetry is induced by the presence of a single magnetic atom within the junction. The single-atom nature of our junctions admits a comprehensive understanding of the observed non-reciprocity, and we find that its origin is qualitatively different from that underlying observations in larger-scale devices. We trace the non-reciprocity of our junctions to dissipation induced by quasiparticle currents flowing in parallel to the supercurrent. In the presence of magnetic atoms, the quasiparticle current can flow by means of YSR subgap resonances, which become asymmetric in the bias direction when particle–hole symmetry is broken. At the relevant junction conductances, the quasiparticle current involves not only direct single-electron tunnelling into the YSR states but also several Andreev reflections exciting the subgap states and thereby contributing to the asymmetry of the quasiparticle current.

Our atomic-scale Josephson junctions provide excellent flexibility for tuning the non-reciprocal behaviour. We have already shown that the magnitude of the asymmetry can be tuned by means of the junction conductance and that the sign of the asymmetry depends on the atomic species inserted into the junction. Considerable opportunities are opened

by combining atomic-scale Josephson junctions with single-atom manipulation. The asymmetry is expected to depend sensitively on the adsorption site of the magnetic atom and can be manipulated by bottom-up creation of atomic assemblies. Thus, our results pave the way towards designing Josephson diodes with a large degree of functional flexibility.

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

## Methods

### Experimental details

The Pb(111) crystal was cleaned by several cycles of Ne$^+$ sputtering and subsequent annealing under ultrahigh-vacuum conditions. Using an electron-beam evaporator, magnetic adatoms (chromium and manganese) were deposited on the clean substrate held at 30 K. The as-prepared sample was then investigated in a CreaTec STM at 1.3 K. The tungsten tip was coated with a sufficiently thick layer of Pb by dipping it into the crystal surface until a full superconducting gap is observed ($\Delta_{tip} = \Delta_{sample}$). Differential-conductance spectra at large junction resistance show the quality of the superconductor–superconductor junction by a superconducting gap of size $2\Delta_{tip} + 2\Delta_{sample} = 4\Delta$ around the Fermi level, flanked by a pair of coherence peaks (grey spectra in Fig. 3a,b).

As Josephson spectroscopy is performed at junction conductances of 20 μS or higher, exceptional tip stability is required to withstand the forces acting at these conductances. Smaller indentations are performed to improve the stability and sharpness of the tip. Individual Pb atoms from the tip apex were deposited by controlled approaches to the flat surface. Measurements were then done on individual Pb, Cr or Mn adatoms on the Pb(111) surface. The Cr and Mn atoms were pulled out from the initial adsorption site by advancing with the STM tip[38].

Josephson spectroscopy was performed by increasing the current set point at a constant bias voltage of 10 mV until reaching the desired junction conductance. After tip stabilization, a large series resistor $R_{series} = 1$ MΩ was introduced into the bias line. This resistance is sufficiently large compared with the junction resistances, so that the junction is effectively current biased. Current-biased Josephson spectroscopy was then performed by sweeping the current bias back and forth between positive and negative values at a rate of 100 nA s$^{-1}$ to 320 nA s$^{-1}$. Datasets with the same tip were recorded at the same ramp rate for direct comparison of magnetic and non-magnetic (Pb) adatoms. Small variations in the ramp rate do not lead to notable changes of the switching and retrapping current. This is in agreement with their logarithmic dependence on the ramp rate. Positive current corresponds to tunnelling of electrons from tip to sample. For statistical analysis, we perform between 500 and 2,000 sweeps in each direction. The STM feedback was turned off during measurement.

We analysed the data using a dedicated Python program. Switching and retrapping events were extracted by taking the derivative of the individual $V$–$I$ curves, which were previously smoothed by a standard Gaussian routine. We also determined $G_{PD}$ from the slope of the $V$–$I$ curve in the trapped state. In analysing the data, we account for several instrumental effects. (1) A slow creep of the piezoelectric elements causes the tip to drift towards the surface, effectively changing the junction conductance. We continuously monitor $G_{PD}$ to characterize the junction and plot all switching and retrapping currents versus $G_{PD}$. (2) The differential amplifier used during the Josephson measurements introduces a slowly shifting voltage offset, which we subtract from the individual $V$–$I$ curves. (3) The voltage/current source has a small offset. For this reason, we correct the entire dataset, including the data measured on the magnetic adatoms, by the mean offset for all recorded data on the pristine Pb–Pb junction under identical measurement conditions, that is, identical tip and identical tip locations. (4) At high junction conductances, the voltage drop across the series resistance of the external circuit becomes non-negligible in the voltage-biased measurements. We correct for this by calibrating the voltage to the superconducting gap size of the Pb–Pb junction.

### Statistical analysis of switching and retrapping currents

As described in the main text, we create a Josephson junction by advancing the STM tip to the surface at a bias voltage (10 mV) far above the superconducting energy gap until the desired normal-state junction conductance (a few tens of μS) is reached. We effectively current bias the junction by inserting a large series resistor ($R_{series} = 1$ MΩ) into the bias line and sweep the current (a few nA) in both directions. The transition from the resistive to the low-resistance state ($I_{re}$) is seen as a sudden drop in the voltage, whereas switching from the low-dissipation to the dissipative branch ($I_{sw}$) occurs as a sudden increase in the voltage. Both events are stochastic in nature, owing to Johnson–Nyquist noise. For this reason, we complement single sweeps by histograms of switching and retrapping currents extracted from a larger set of $V$–$I$ curves. Non-reciprocity of the switching and retrapping current is then seen as asymmetries between the histograms for positive and negative bias. Extended Data Fig. 1 shows corresponding histograms extracted from 500 to 2,000 sweeps recorded on Pb, Cr and Mn junctions with $G_N$ equal to 50 μS. For the Pb junction, the histograms of the switching currents $|I_{sw,+}|$ and $|I_{sw,-}|$ exhibit broad Gaussian-like distributions, with the same average ((5.9 ± 0.4) nA) for both bias directions. The histograms of the retrapping currents $|I_{re,+}|$ and $|I_{re,-}|$ are narrower ((1.8 ± 0.1) nA) but also independent of bias direction. The histograms for Cr and Mn junctions are qualitatively different. The histograms of the retrapping currents exhibit a clear relative shift between the two bias directions, leading to different absolute values of the averages of $I_{re,+}$ ((1.9 ± 0.2) nA for Cr and (1.86 ± 0.04) nA for Mn) and $I_{re,-}$ ((−1.4 ± 0.2) nA for Cr and (−2.18 ± 0.06) nA for Mn). The histograms of the switching current exhibit a noticeable but much weaker dependence on the bias direction.

### Analysis of switching and retrapping currents as a function of $G_{PD}$

The histograms in Extended Data Fig. 1 reflect the stochastic nature of the switching and retrapping processes but are further broadened by piezoelectric creep over the course of the measurement. The creep effectively increases the junction conductance (as quantified by the phase-diffusion conductance $G_{PD}$) with time. For each, we minimize the creep-induced broadening by using 100 consecutive sweeps for separate histograms with an associated average $G_{PD}$. Extended Data Figs. 2–4 illustrate this analysis. The histogram for the earliest 100 sweeps are shown at the bottom of each panel. Histograms obtained from subsequent batches of 100 sweeps correspond to larger junction conductances $G_{PD}$, as indicated in the figures. This increase is accompanied by an increase in $|I_{sw}|$ and $|I_{re}|$ as seen by a shift of the corresponding histograms. This scheme is the basis for Fig. 2, which collects the average retrapping currents, along with the standard deviations of all of these histograms.

### Comparison of switching currents

Extended Data Fig. 5 shows the switching currents of Cr and Mn junctions as a function of $G_{PD}$, in both cases compared with Pb junctions measured with the same tip. For identical tips, the switching currents $|I_{sw}|$ show almost the same linear dependence on $G_{PD}$, justifying the use of $G_{PD}$ as a suitable measure of junction conductance.

### Influence of the STM tip

The STM tip is an integral part of our atomic-scale Josephson junctions. To ensure that the main findings remain valid independent of details of the tip apex, we investigated several tips obtained through reshaping by large tip indentations into the Pb substrate. Extended Data Fig. 6 shows the non-reciprocity of the retrapping current as a function of $G_{PD}$ for junctions including Cr and Mn atoms but measured with different tips. All tips show a positive value of the asymmetry $\Delta I_{re} = |I_{re,+}| - |I_{re,-}|$ in case of Cr, a negative value for Mn and no asymmetry for Pb adatoms. Although these qualitative observations are robust for all tips, there are small differences in the magnitude of the asymmetry at the same value of $G_{PD}$. We tentatively ascribe these variations to tip-dependent Josephson coupling energies and quasiparticle currents, as well as noise levels.

## YSR states and particle–hole symmetry

We trace the asymmetric quasiparticle current to YSR states, which the magnetic impurity induces within the superconducting gap of the substrate superconductor. Here we briefly explain the origin of this asymmetry and its relation to breaking of particle–hole symmetry. The interaction of the magnetic impurity (impurity spin $\mathbf{S}$) with the conduction electrons of the substrate (field operators $\psi_{\mathbf{k},\sigma}$ with wavevector $\mathbf{k}$ and spin $\sigma$) takes the form

$$H_{\text{int}} = \sum_{\mathbf{k},\mathbf{k}'} \sum_{\sigma,\sigma'} \psi^{\dagger}_{\mathbf{k},\sigma} [J\mathbf{S} \cdot \mathbf{s}_{\sigma,\sigma'} + K\delta_{\sigma,\sigma'}]\psi_{\mathbf{k}',\sigma'}. \tag{2}$$

Here $\mathbf{s} = \frac{1}{2}\boldsymbol{\sigma}$ with the vector of Pauli matrices $\boldsymbol{\sigma}$, $J$ denotes the strength of the exchange coupling and $K$ the strength of potential scattering. Focusing for simplicity on spin-$\frac{1}{2}$ impurities, this interaction can be obtained from the Anderson impurity model (impurity level with energy $\epsilon < 0$, on-site interaction $U > 0$, hybridization $t$) by a Schrieffer–Wolff transformation[43]. This yields

$$J = |t|^2 \left\{ \frac{1}{|\epsilon|} + \frac{1}{\epsilon + U} \right\}, \quad K = |t|^2 \left\{ \frac{1}{|\epsilon|} - \frac{1}{\epsilon + U} \right\}. \tag{3}$$

In the particle–hole symmetric situation $\epsilon = -\frac{U}{2}$, the excitation energies are identical for the empty configuration ($|\epsilon|$) and the doubly occupied configuration ($\epsilon + U$). In this case, the potential scattering $K$ vanishes. Potential scattering becomes non-zero when the empty and doubly occupied configurations have different excitation energies, reflecting broken particle–hole symmetry. The sign of $K$ depends on which of the configurations has the higher excitation energy.

A standard calculation[44–47] shows that the YSR state induced by the magnetic adatom induces a pair of subgap states whenever the exchange coupling $J$ is non-zero. For $K = 0$, the electron wavefunction $u$ and the hole wavefunction $v$ of the bound state are equal to each other, as expected in a particle–hole symmetric situation. An asymmetry between the electron and hole wavefunctions appears when potential scattering is non-zero. The sign of the asymmetry depends on the sign of $K$.

This asymmetry between the electron and hole wavefunctions explains the asymmetric current–voltage characteristic of junctions including a magnetic impurity. Single-electron tunnelling at positive (negative) bias is proportional to $|u|^2$ ($|v|^2$), in which the wavefunctions are evaluated at the tip position (Fermi's golden rule). Similarly, several Andreev reflections will also involve these factors when the final state involves an excitation of the subgap YSR state. As particle–hole symmetry requires fine tuning of the impurity parameters, one generically expects that the current–voltage characteristics of junctions involving a magnetic adatom are asymmetric. The direction of the asymmetry depends on the details of the atomic physics of the adatom, consistent with our observation of opposite asymmetries for Mn and Cr.

We note that these considerations are independent of whether time-reversal symmetry is broken or not. The asymmetry of the YSR wavefunctions is controlled by potential scattering—and thus only by breaking of particle–hole symmetry—even when the adatom spin is polarized and time-reversal symmetry is explicitly broken[44–47]. Of course, broken time-reversal symmetry may lead to asymmetries in the current–phase relation as well, which might induce a coexisting non-reciprocity of the switching current.

In the absence of thermal fluctuations, the switching and retrapping currents can be obtained from the junction dynamics as follows. In the absence of fluctuations, the junction switches at the critical current, that is, at the current bias at which the tilt of the washboard potential eliminates the minima. In this limit, an asymmetry in the switching current requires an asymmetric washboard potential or, equivalently, an asymmetric current–phase relation. Fluctuations will then reduce the switching current below the critical current but, in the limit of weak damping (pronounced hysteresis), the asymmetry of the switching current is largely inherited from the asymmetry in the critical current.

On the other hand, the retrapping current is the result of different physics. In the absence of fluctuations and at weak damping, the junction retraps, once the energy gain owing to the bias current (that is, owing to the tilt in the language of the washboard potential) becomes smaller than the frictional energy loss during the motion. The energy gain depends on the tilt but not on the shape (or the asymmetry) of the washboard potential. Thus, an asymmetry can only arise from asymmetries in the frictional energy loss, which is associated with the quasiparticle current at the microscopic level. Fluctuations tend to increase the retrapping current but the asymmetry is essentially inherited from the asymmetry in the retrapping currents of the junction in the absence of fluctuations.

## Simulations of the RCSJ model

Our theoretical simulations underlying Fig. 4 are based on the RCSJ model for a current-biased junction[28,29],

$$I_{\text{bias}} = C\frac{\text{d}}{\text{d}t}V + I_s(\varphi) + I_d(V) + \delta I. \tag{4}$$

Here $I_{\text{bias}}$ is the current bias, $V$ the voltage drop at the junction, $C$ the junction capacitance and $\varphi$ the phase difference across the junction. We assume a symmetric and sinusoidal current–phase relation $I_s(\varphi) = I_c \sin\varphi$. We allow for a general nonlinear dissipative current $I_d(V)$, with associated Nyquist noise $\delta I$ with correlator $\langle \delta I(t)\delta I(t') \rangle \propto \delta(t - t')$ (see below). When combined with the Josephson relation $V = (\hbar/2e)$ $\text{d}\varphi/\text{d}t$, equation (4) gives a Langevin equation for the phase difference across the junction. We solve the Langevin equation by Monte Carlo integration, accounting for the current sweep, to obtain the results shown in Fig. 4 (with further details in Extended Data Fig. 7), as well as in Extended Data Figs. 8 and 9.

The dissipative current $I_d(V)$ includes the quasiparticle current $I_{\text{qp}}(V)$, which we extract from experimental $I$–$V$ traces (see below for details). To account for the observed phase diffusion in the trapped state, we also incorporate frequency-dependent friction. Following Kautz and Martinis[30], we shunt the junction by an extra $RC$ element with ohmic resistor $\widetilde{R}$ and capacitor $\widetilde{C}$ to model dissipation induced by the electromagnetic environment. The total dissipative current is then the sum of the quasiparticle current and the current flowing by means of the $RC$ element,

$$I_d(V) = I_{\text{qp}}(V) + \frac{V - \widetilde{V}}{\widetilde{R}}, \quad \delta I = \delta I_{\text{qp}} + \delta I_{\widetilde{R}}, \tag{5}$$

in which $\widetilde{V}$ is the voltage drop across the capacitor, which satisfies the equation

$$\frac{\text{d}}{\text{d}t}\widetilde{V} = \frac{1}{\widetilde{R}\widetilde{C}}(V - \widetilde{V} + \widetilde{R}\delta I_{\widetilde{R}}). \tag{6}$$

The $RC$ element is inconsequential at low frequencies (running state), so that damping is dominated by the quasiparticle current. By contrast, it dominates friction at high frequencies (trapped state), allowing for phase diffusion. We assume $V/\widetilde{R} \gg I_{\text{qp}}(V)$, so that the quasiparticle current is effectively shorted at high frequencies, $I_d(V) \simeq V/\widetilde{R}$. The Nyquist noise associated with the quasiparticle current has correlator $\langle \delta I_{\text{qp}}(t)\delta I_{\text{qp}}(t') \rangle = 2T [I_{\text{qp}}(V)/V]\delta(t - t')$, whereas the Nyquist noise associated with the resistor $\widetilde{R}$ has correlator $\langle \delta I_{\widetilde{R}}(t)\delta I_{\widetilde{R}}(t') \rangle = 2T\widetilde{R}^{-1}\delta(t - t')$.

Measuring time in units of the inverse plasma frequency, $\tau = \omega_p t$ with $\omega_p = [2eI_c/\hbar C]^{1/2}$, and currents in units of the critical current, $i = I/I_c$, the resulting RCSJ equations become

$$\frac{\mathrm{d}}{\mathrm{d}\tau}\varphi = \upsilon,$$

$$\frac{\mathrm{d}}{\mathrm{d}\tau}\upsilon = i_\mathrm{b} - i_\mathrm{s}(\varphi) - \left[i_\mathrm{qp}(\upsilon) + \frac{\upsilon - \tilde{\upsilon}}{\tilde{Q}}\right] \quad (7)$$
$$- \sqrt{2\theta[i_\mathrm{qp}(\upsilon)/\upsilon]}\,\xi_1 - \sqrt{2\tilde{\theta}/\tilde{Q}}\,\xi_2,$$

$$\frac{\mathrm{d}}{\mathrm{d}\tau}\tilde{\upsilon} = \frac{1}{\tilde{\tau}}(\upsilon - \tilde{\upsilon} + \sqrt{2\tilde{\theta}\tilde{Q}}\,\xi_2),$$

in which we defined the dimensionless voltages $\upsilon = 2eV/\hbar\omega_\mathrm{p}$ and $\tilde{\upsilon} = 2e\tilde{V}/\hbar\omega_\mathrm{p}$, the dimensionless currents $i_\mathrm{b} = I_\mathrm{bias}/I_\mathrm{c}$, $i_\mathrm{s}(\varphi) = I_\mathrm{s}(\varphi)/I_\mathrm{c} = \sin\varphi$ and $i_\mathrm{qp}(\upsilon) = I_\mathrm{qp}(\hbar\omega_\mathrm{p}\upsilon/2e)/I_\mathrm{c}$, the effective quality factor $\tilde{Q} = \tilde{R}C\omega_\mathrm{p}$ at large frequencies, as well as the reduced temperatures $\theta = T/E_\mathrm{J}$ and $\tilde{\theta} = \tilde{T}/E_\mathrm{J}$. (Here $E_\mathrm{J} = \hbar I_\mathrm{c}/2e$ is the Josephson energy and $\tilde{T}$ is the temperature of the resistor $\tilde{R}$). We also defined dimensionless Langevin currents $\xi_1$ and $\xi_2$ with normalized correlations $\langle \xi_i(\tau)\xi_j(\tau')\rangle = \delta_{ij}\delta(\tau - \tau')$ corresponding to $\delta I_\mathrm{qp}$ and $\delta I_{\tilde{R}}$, respectively. We estimate the experimental parameters as $R_\mathrm{N} \approx 20\,\mathrm{k\Omega}$, $\Delta \approx 1.5\,\mathrm{meV}$, $T \approx 0.1\,\mathrm{meV}$ and $C \approx 10^{-15}\,\mathrm{F}$. This gives $I_\mathrm{c} \approx 100\,\mathrm{nA}$, $E_\mathrm{J} \approx 0.2\,\mathrm{meV}$ and $\hbar\omega_\mathrm{p} \approx 0.3\,\mathrm{meV}$. The reduced temperature is thus $\theta = 0.5$. For the $RC$ element, we choose parameters $\tilde{Q} = 10$, $\tilde{\tau} = 1{,}000$ and $\tilde{\theta} = \theta$. We sweep the bias current with a rate $\mathrm{d}I_\mathrm{bias}/\mathrm{d}t = 10^{-7}I_\mathrm{c}\omega_\mathrm{p} \approx 1\,\mathrm{nA}\,\mathrm{\mu s}^{-1}$. The experimental sweep rate is smaller by a factor of about $10^{-3}$ but this would make the numerical simulations forbidding. Along with the simplified current–phase relation and the order-of-magnitude estimates of experimental parameters, this implies that one can only expect qualitative but not quantitative agreement between simulations and experiment.

Theoretical simulations of single traces are shown in Fig. 4. The corresponding close-up view of the absolute values of the currents in Extended Data Fig. 7 brings out the large asymmetry of the retrapping current and the smaller (and, according to the histograms, largely statistical) asymmetry of the switching current. In Extended Data Fig. 8, we show the histograms of the absolute values of switching and retrapping currents extracted from 100 sweeps in each current direction. Note that the panels only differ in the precise form of $I_\mathrm{qp}(V)$, which is extracted from the $I$–$V$ curves of Pb, Cr and Mn, respectively. The simulations based on the $I_\mathrm{qp}(V)$ of Pb do not show asymmetry in the switching or the retrapping currents. The simulations based on the $I_\mathrm{qp}(V)$ of Cr and Mn exhibit weak asymmetry in the switching currents and strong asymmetry in the retrapping currents, correctly reproducing the qualitative features of the experimental histograms in Extended Data Figs. 2–4.

## Asymmetric current–phase relation

To rule out the possibility that the observed asymmetry stems from the current–phase relation $I_\mathrm{s}(\varphi)$ rather than from the dissipative quasiparticle current, we now demonstrate that an asymmetric current–phase relation leads to strong asymmetry in the switching currents and weak asymmetry in the retrapping currents, in contrast to our experimental observations. To this end, we simulate equation (7) using Pb $I$–$V$ data for $I_\mathrm{qp}(V)$ together with an asymmetric current–phase relation

$$I_\mathrm{s}(\varphi) = I_0[\sin(\varphi - \varphi_0) + b\sin(2\varphi)]. \quad (8)$$

We choose $\varphi_0 = 0.5 = b$ and fix $I_0 \simeq 54.2\,\mathrm{nA}$ by requiring that the current entering the definition of the plasma frequency, that is, the slope of $I_\mathrm{s}$ around the stable minimum, is still $100\,\mathrm{nA}$ (which we continue to use as the unit of current). The critical current now depends on direction, with $I_{\mathrm{c},+} \simeq 53.3\,\mathrm{nA}$ and $I_{\mathrm{c},-} \simeq 80.0\,\mathrm{nA}$. Histograms of switching and retrapping currents obtained by simulating equation (7) with the current–phase relation given in equation (8) are presented in Extended Data Fig. 9. The asymmetry of the switching currents is clearly much greater than that of the retrapping currents. Thus, a symmetric dissipative current together with an asymmetric current–phase relation cannot explain the phenomenology of strongly asymmetric retrapping currents and weakly asymmetric switching currents observed for the Cr and Mn Josephson junctions.

## Extraction of quasiparticle current

We extract the quasiparticle contribution to the dissipative current $I_\mathrm{qp}(V)$ from voltage-biased measurements of Pb, Cr and Mn junctions at the normal-state conductance of $G_\mathrm{N} = 50\,\mathrm{\mu S}$ (see Fig. 3f). As well as the quasiparticle current, these traces include a Josephson peak originating from incoherent Cooper-pair tunnelling. We remove the Josephson contribution $I_\mathrm{J}(V)$ by fitting to the phenomenological expressions[48]

$$I_\mathrm{meas}(V) = I_\mathrm{J}(V + V_\mathrm{offset}) + I_{\mathrm{qp},0}(V + V_\mathrm{offset}) + I_\mathrm{offset}, \quad (9\mathrm{a})$$

$$I_\mathrm{J}(V) = A\frac{V\delta V}{V^2 + \delta V^2} + B\frac{V^3\delta V}{(V^2 + \delta V^2)^2}, \quad (9\mathrm{b})$$

$$I_{\mathrm{qp},0}(V) = CV + DV^2 + EV^3, \quad (9\mathrm{c})$$

over a voltage range $e|V| \ll \Delta$, which contains the Josephson peak. (We chose $e|V| < 0.32\,\mathrm{meV}$). We also account for offsets in the measured voltage and current through the parameters $V_\mathrm{offset}$ and $I_\mathrm{offset}$. The fit parameters are collected in Extended Data Table 1. We then subtract the Josephson contribution as well as the offsets from the measured data to isolate the quasiparticle contribution. To reduce the fluctuations at small $V$ associated with the Josephson contribution, a Gaussian filter (width $\sigma = 5$ data points $\simeq 0.55\,\mathrm{mV}$) is applied to the isolated quasiparticle current data. Finally, $I_\mathrm{qp}(V)$ is obtained by interpolation using a linear splining procedure, enforcing $I_\mathrm{qp}(0) = 0$.

## Data availability

The data that support the findings of this study have been deposited in the Refubium database: https://doi.org/10.17169/refubium-37060.

## Code availability

The codes used for data analysis and simulations have been deposited in the Refubium database: https://doi.org/10.17169/refubium-37060.

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

**Acknowledgements** We thank K. Biel for assistance in preliminary measurements and C. Lotze for general technical support. We acknowledge financial support by the Deutsche Forschungsgemeinschaft (DFG, German Research Foundation) through projects 277101999 (CRC 183, project C03), FR2726/5 and SFB 910 (project A11), as well as by Agence Nationale de la Recherche under grant JOSPEC.

**Author contributions** M.T. carried out the experiments, with the assistance of B.M., I.T., G.R. and C.B.W. O.P. and N.B. set up the Josephson circuit with the help of C.B.W. and carried out preliminary measurements. L.M. and J.F.S. contributed theoretical considerations and the model calculations. M.T., L.M., J.F.S., I.T., C.B.W., F.v.O. and K.J.F. interpreted the data. K.J.F. conceived the experiment, with the assistance of C.B.W. K.J.F. guided the experiment. F.v.O. conceived and guided the theory. M.T., F.v.O. and K.J.F. wrote the paper, with input from all coauthors.

**Funding** Open access funding provided by Freie Universität Berlin.

**Competing interests** The authors declare no competing interests.

**Additional information**
**Correspondence and requests for materials** should be addressed to Katharina J. Franke.

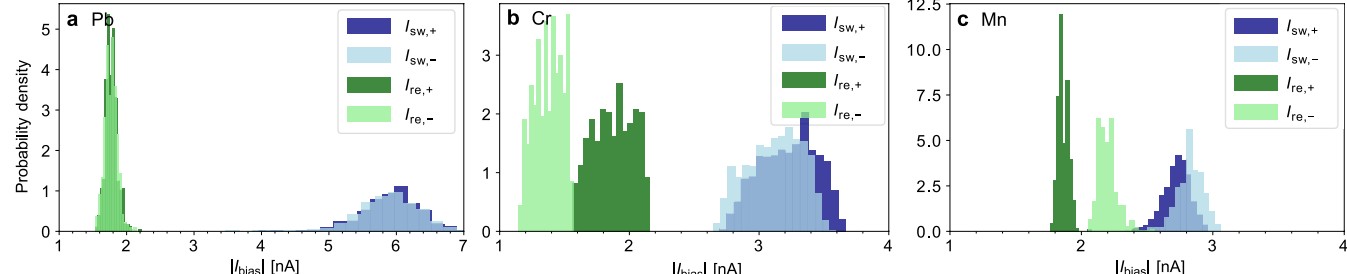

**Extended Data Fig. 1 | Statistics of switching and retrapping currents for single-atom Josephson junctions. a–c**, Histograms of absolute values of switching and retrapping currents for the two bias directions, as extracted from individual *V–I* curves for Pb (**a**), Cr (**b**) and Mn (**c**) junctions. The histograms in **a** and **c** include data extracted from 500 sweeps and **b** includes 2,000 sweeps for each current direction. The junction conductances $G_N$ were set at 10 mV to 50 μS. The distributions of switching and retrapping currents arise from the stochastic nature of switching and retrapping events, and are further broadened by piezoelectric creep while taking the 500 to 2,000 sweeps (see Extended Data Figs. 2–4 for histograms without this extra broadening).

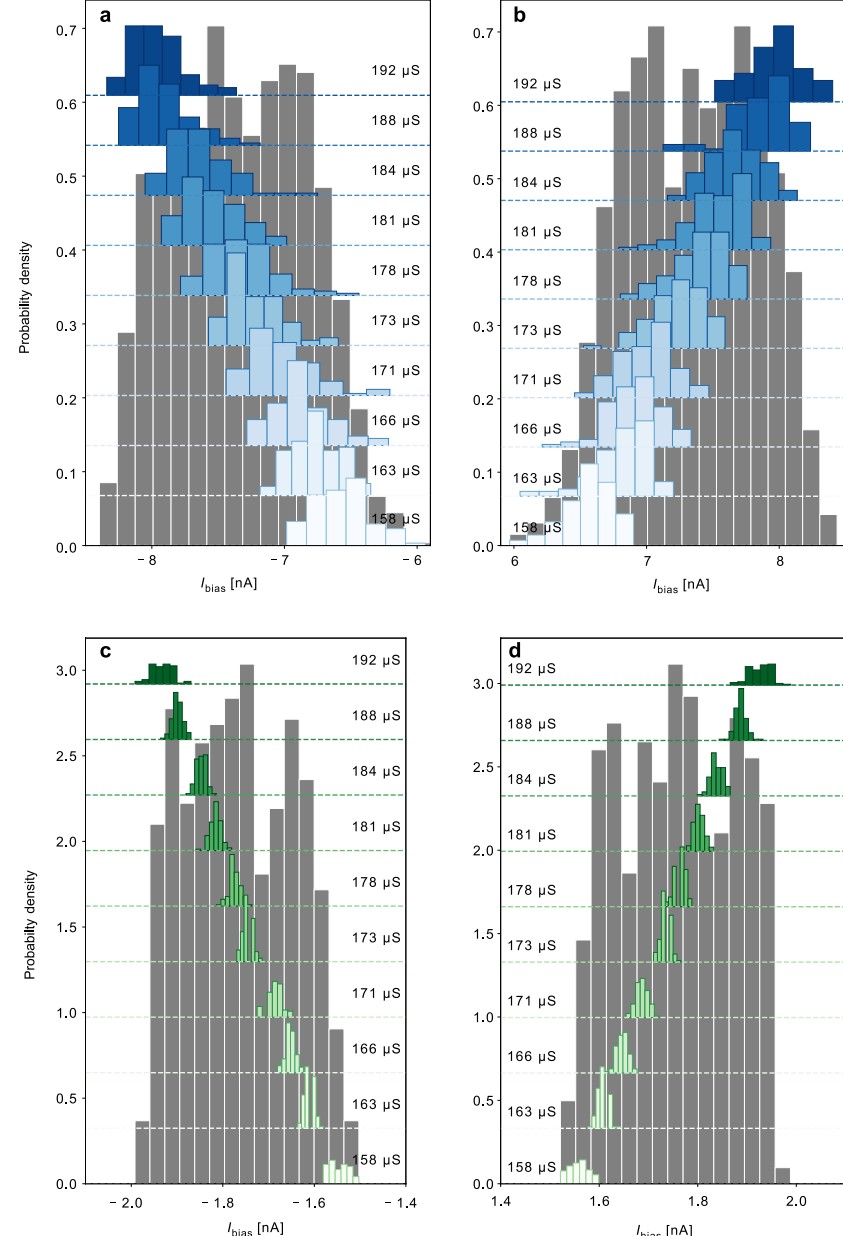

**Extended Data Fig. 2 | Evolution of histograms with $G_{PD}$ for a Pb junction.** The grey histograms (background) are extracted for switching (**a**,**b**) and retrapping (**c**,**d**) currents from 2,000 individual *V–I* curves, recorded after setting the junction to a (high-voltage) conductance of $G_N = 50$ µS. The positive-bias (**a**) and negative-bias (**b**) switching currents were divided into bins of 100 sweeps each (blue histograms). The same procedure was implemented for positive-bias (**c**) and negative-bias (**d**) retrapping currents (green histograms). Every other histogram is omitted for clarity. $G_{PD}$ varies owing to piezoelectric drift. Its average value is indicated for each of the histograms. The piezoelectric drift to larger $G_{PD}$ over the course of the measurement is reflected in shifts to higher absolute values of switching and retrapping currents. Note that these data were recorded with a different tip to those in Extended Data Fig. 1.

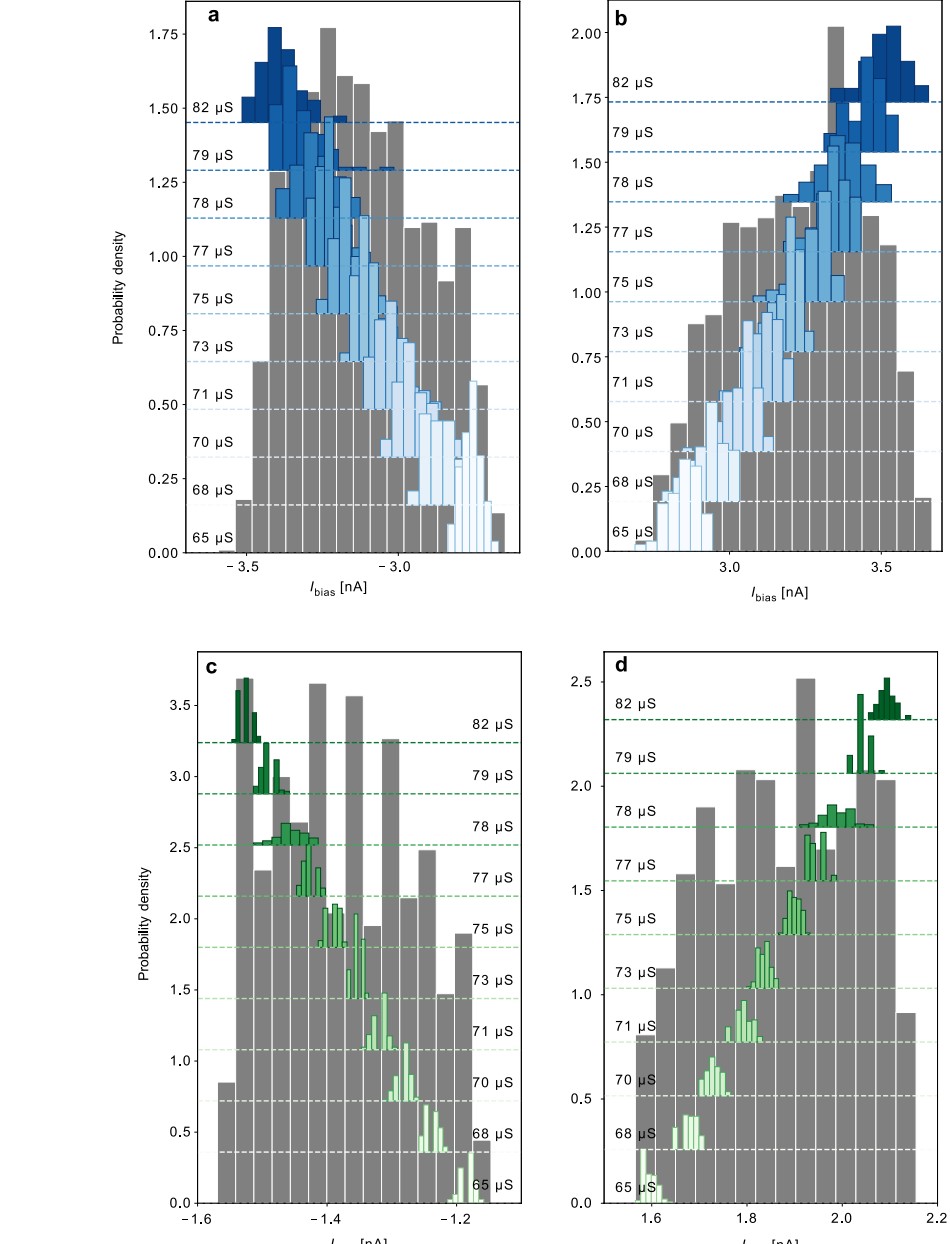

**Extended Data Fig. 3 | Evolution of histograms with $G_{PD}$ for a Cr junction.** The grey histograms (background) are extracted for switching (**a**,**b**) and retrapping (**c**,**d**) currents from 2,000 individual $V$–$I$ curves, recorded after setting the junction to a (high-voltage) conductance of $G_N = 50$ µS. The positive-bias (**a**) and negative-bias (**b**) switching currents were divided into bins of 100 sweeps each (blue histograms). The same procedure was implemented for positive-bias (**c**) and negative-bias (**d**) retrapping currents (green histograms). Every other histogram is omitted for clarity. $G_{PD}$ varies owing to piezoelectric drift. Its average value is indicated for each of the histograms. The piezoelectric drift to larger $G_{PD}$ over the course of the measurement is reflected in shifts to higher absolute values of switching and retrapping currents.

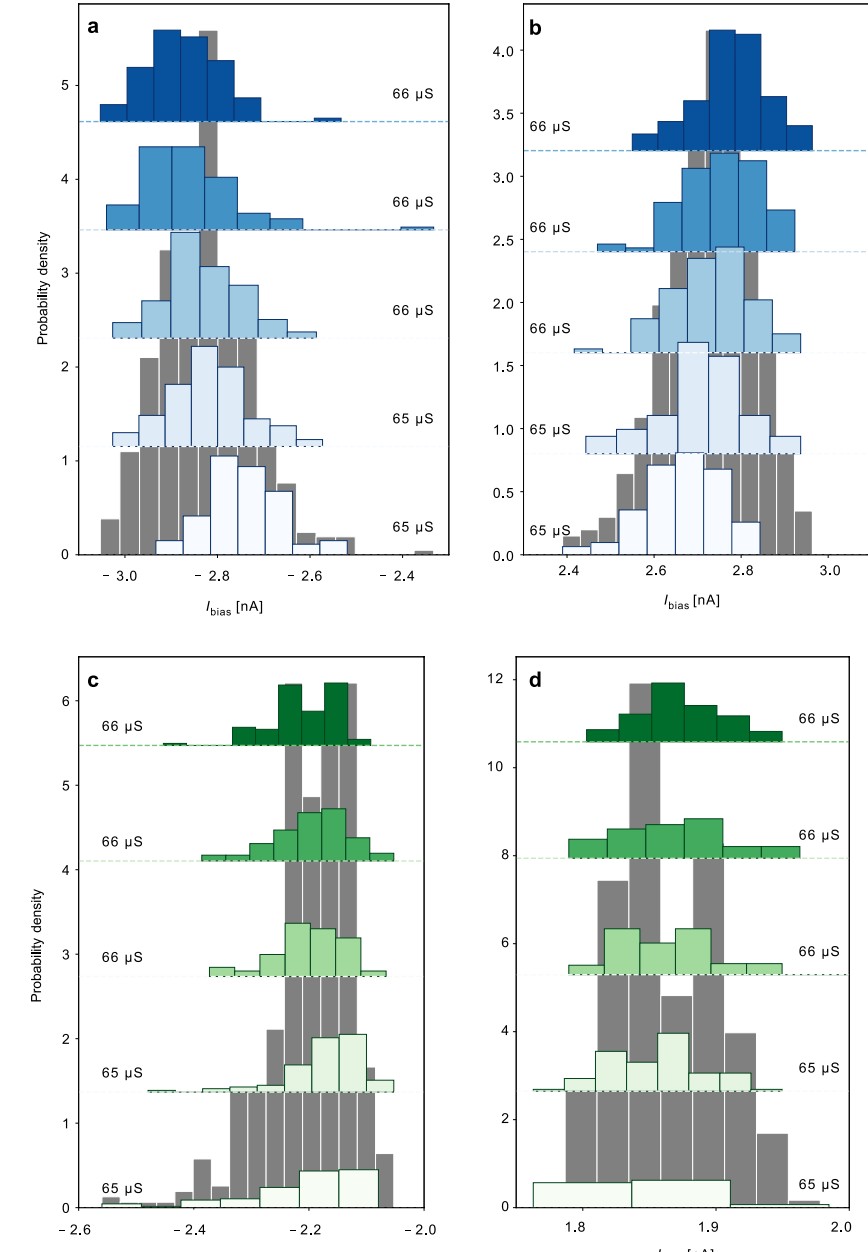

**Extended Data Fig. 4 | Evolution of histograms with $G_{PD}$ for a Mn junction.** The grey histograms (background) are extracted for switching (**a**,**b**) and retrapping (**c**,**d**) currents from 500 individual $V$–$I$ curves, recorded after setting the junction to a (high-voltage) conductance of $G_N = 50$ μS. The positive-bias (**a**) and negative-bias (**b**) switching currents were divided into bins of 100 sweeps each (blue histograms). The same procedure was implemented for positive-bias (**c**) and negative-bias (**d**) retrapping currents (green histograms). $G_{PD}$ varies owing to piezoelectric drift. Its average value is indicated for each of the histograms. The piezoelectric drift to larger $G_{PD}$ as well as the shifts to higher absolute values of switching and retrapping currents are less pronounced than in Extended Data Fig. 2, as the time of measurement was much shorter.

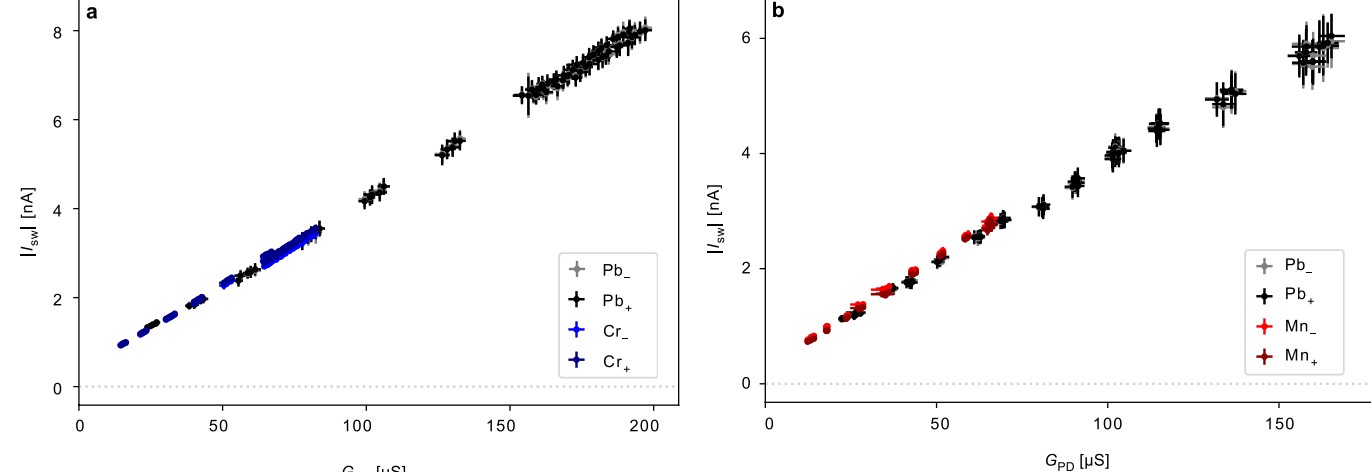

**Extended Data Fig. 5 | Comparison of switching currents for the Cr and Mn junctions with the reference data for Pb junctions. a**, Extracted positive and negative switching currents for Cr and Pb junctions with normal-state conductances $G_N$ between 20 and 50 μS. The data were acquired with the same tip and under similar measurement conditions. **b**, Extracted positive and negative switching currents for Mn and Pb junctions with normal-state conductances $G_N$ between 20 and 50 μS. The switching current depends linearly on $G_{PD}$, with the same slope for magnetic and non-magnetic atoms, provided data are taken under corresponding measurement conditions.

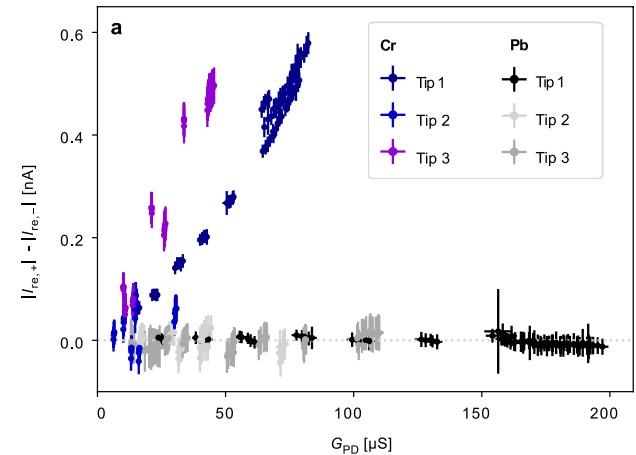

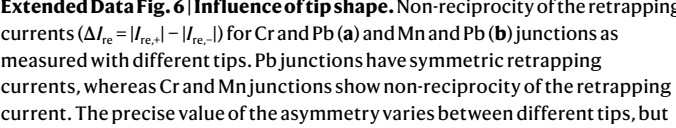

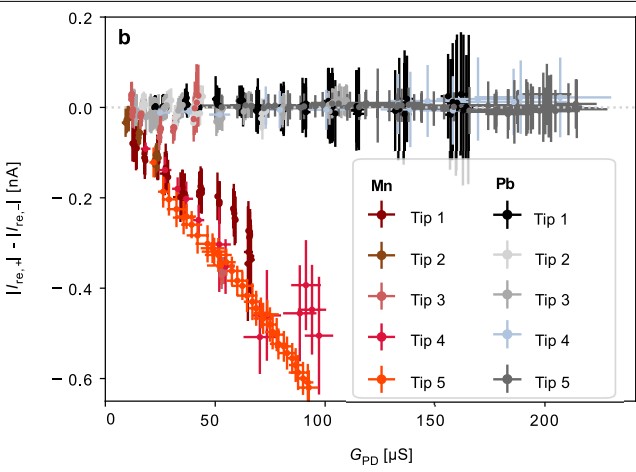

**Extended Data Fig. 6 | Influence of tip shape.** Non-reciprocity of the retrapping currents ($\Delta I_{re} = |I_{re,+}| - |I_{re,-}|$) for Cr and Pb (**a**) and Mn and Pb (**b**) junctions as measured with different tips. Pb junctions have symmetric retrapping currents, whereas Cr and Mn junctions show non-reciprocity of the retrapping current. The precise value of the asymmetry varies between different tips, but the sign of the asymmetry is consistently opposite for Cr and Mn. The (high-voltage) junction conductances $G_N$ were set between 20 and 50 µS at 10 mV. $G_{PD}$ was determined from individual $V$–$I$ sweeps as described in Methods. The asymmetry was derived from $I_{sw}$ and $I_{re}$ after averaging over 100 sweeps.

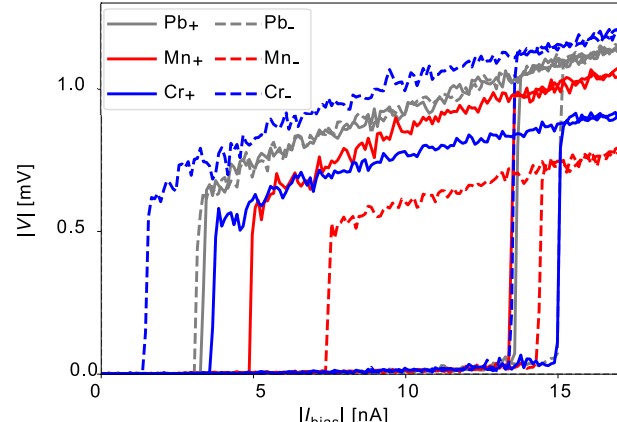

**Extended Data Fig. 7 | Modelling of the non-reciprocity within the RCSJ model.** $|V| - |I|$ traces corresponding to the simulations in Fig. 4. This close-up view brings out the strong asymmetry of the retrapping current and includes the asymmetry of the switching currents. Traces are shown for quasiparticle currents extracted from measurements on Pb (grey), Cr (blue) and Mn (red). As shown by the histograms in Extended Data Fig. 8, the asymmetry in the switching current is largely owing to statistical fluctuations between different current ramps. Thus, the underlying non-reciprocity in the switching current is actually considerably smaller than the asymmetry shown in this particular trace.

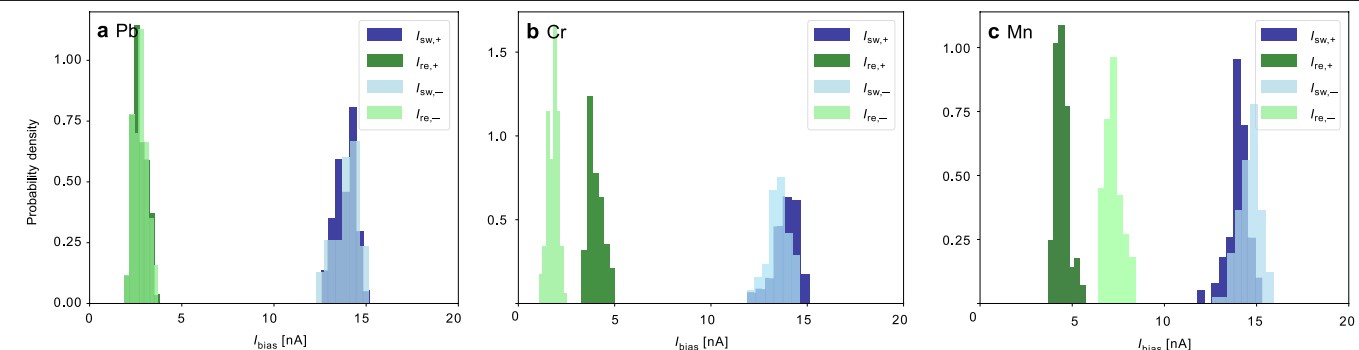

**Extended Data Fig. 8 | Statistics of switching and retrapping currents (theoretical simulations based on *I–V* measurements). a–c,** Histograms of absolute values of switching and retrapping currents for the two bias directions, as extracted from individual *V–I* curves from simulation of equation (7) with $I_{qp}(V)$ obtained from experimental *I–V* curves of a Pb (**a**), Cr (**b**) and Mn (**c**) junction at $G_N$ = 50 μS (compare Fig. 3f as well as equation (9) and corresponding text). Each histogram includes data extracted from 100 sweeps for each current direction. For parameters, see text below equation (7).

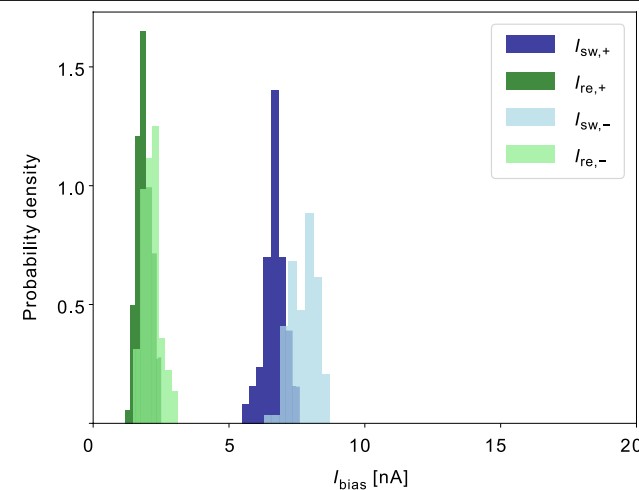

**Extended Data Fig. 9 | Simulated statistics of switching and retrapping currents with symmetric quasiparticle current and asymmetric current–phase relation.** Histograms of absolute values of switching and retrapping currents for the two bias directions, as extracted from individual $V$–$I$ curves in simulations of equation (7). $I_{qp}(V)$ is obtained from experimental $I$–$V$ curves of a Pb junction at $G_N = 50$ µS. The asymmetric current–phase relation is given in equation (8). Each histogram includes data extracted from 98 sweeps for each current direction. Other parameters as in Extended Data Fig. 8.

**Extended Data Table 1 | Quasiparticle current fitting parameters**

|  | $V_{offset}$ [mV] | $I_{offset}$ [nA] | $\delta V$ [mV] | $A$ [nA] | $B$ [nA] | $C$ [nA/mV] | $D$ [nA/mV$^2$] | $E$ [nA/mV$^3$] |
|---|---|---|---|---|---|---|---|---|
| Pb | 0.0187 | 0.0292 | 0.135 | 16.4 | -20.6 | 7.00 | 0.121 | -21.8 |
| Cr | 0.0210 | 0.00169 | 0.140 | 5.96 | -7.71 | 4.01 | 2.47 | -1.01 |
| Mn | 0.0129 | -0.123 | 0.138 | 8.47 | -10.7 | 3.66 | -2.52 | 15.1 |

Fitting parameters for extracting the quasiparticle current $I_{qp}(V)$ from the measured current $I_{meas}(V)$ by subtracting the Josephson peak owing to incoherent Cooper-pair tunnelling (see equation (9) and corresponding text).