## [Peer Review File · Nature]

Manuscript Title: Diode effect in Josephson junctions with a single magnetic atom

Reviewer Comments & Author Rebuttals

Reviewer Reports on the Initial Version:

Referees' comments:

Referee #1 (Remarks to the Author):

Using scanning tunneling spectroscopy, the authors report observation of diode effect in Josephson junctions that contain single magnetic atoms. In the current biased spectroscopy measurements, they observe non-reciprocal behavior in both switching as well as retrapping currents, however, non-reciprocity is stronger in the retrapping current which is different compared to recent diode effects observed in other systems. While non-reciprocity in other systems has been ascribed to the breaking of inversion and time-reversal symmetry, authors claim that the observed non-reciprocity is due to sub-gap states resulting from the interaction between magnetic atoms with the substrate which drives asymmetric damping of the phase particle during retrap. Their claim is based on an extended RCSJ model that includes quasiparticle current via sub-gap states due to the magnetic impurity and they also exclude asymmetric current-phase relation which mainly affects the switching current. In view that the superconducting diode effect has recently gained interest due to its potential in realizing high-speed electronics without dissipation of the currents, this manuscript stimulates additional ways to realize the superconducting diode effect. Therefore, I recommend considering publication in Nature after the following concerns are clarified.

- 1) What are the dark circles closeby the magnetic atoms in the inset image of Fig. 1a. Can authors rule out their role in the observed spectra?
- 2) All junctions show phase diffusion conductance which leads to finite dissipation. Are there ways to improve the quality of junctions to minimize/remove phase slip events? How do current biased spectra look on bare Pb substrate? Does the finite size of the tip-superconductor induce additional contribution to the G_{pd} ? The last one can be checked by measuring the width of the Josephson peak and seeing if it is broadened by the temperature alone.
- 3) The authors mention about biasing junctions with smaller rates during the spectroscopy. Can authors comment on the response of the junctions with faster switching and if/how the finite G_{pd} can limit the switching speed?
- 4) Authors quoted in the manuscript (line 195) and supplementary (page 7) that simulations for Pb do not show asymmetry in the switching/retrapping currents. However, Fig 4 clearly shows a large asymmetry in the switching. Can the authors clarify the discrepancy in the simulation?

In addition, authors can consider the following optional comments.

- 1) Owing to the large asymmetry in the observed currents, authors can consider quantifying the diode efficiency.
- 2) What happens to the junction characteristics when the time-reversal symmetry is broken (if they

have access to in-plane magnetic fields)?

3) Did authors try similar spectroscopy off the atoms on bare Pb in the vicinity of magnetic atoms and especially close to the dark impurities?

4) There is an error at the beginning of line 265.

Referee #2 (Remarks to the Author):

The authors use an STM to study Josephson junctions through an individual magnetic adatom. They observe an asymmetry of the retrapping currents that depends on the type of adatom. The switching currents display a much weaker asymmetry. The results are presented in the context of so-called Josephson diodes that have received a lot of attention recently. While the asymmetry in these Josephson diodes arises due to an interplay of broken inversion symmetry and broken time-reversal symmetry, the authors attribute their observations to a different mechanism, namely an interplay between broken inversion symmetry and broken particle-hole symmetry. While the energy spectrum of the junction is particle-hole symmetric, the wavefunctions of the YSR states forming in the junctions are not, leading to different intensities in tunneling spectroscopy at positive and negative bias - an effect that is well known, but to my knowledge has not been linked to asymmetries in switching/retrapping currents before. The authors back up their claim using numerical simulations based on the RCSJ-model and taking an asymmetric current-voltage characteristic as an input.

The results are interesting and very timely. To recommend them for publication in Nature, the main message would have to be clarified though.

While the paper is well written, I find the abstract and introduction somewhat misleading: they suggest that what the authors achieved is a miniaturization of the Josephson diode as discussed in earlier works - before then evoking a completely different mechanism. The work should be set in a broader context and mention other possible reasons for asymmetries (such as self-field effects). The conclusion as well stresses the miniaturization aspect, even though it doesn't seem to me that the difference in mechanism is due to the size of the junction. It would be useful to distinguish (or relate more clearly) the two aspects.

Some more specific comments/questions:

- Fig. 1:

For some values of G_{PD} (above 60 μS), there are two different data points. Why?

Furthermore, for the clarity of the figure, it might be better to use a larger contrast between dark and bright colors.

- Fig. 3:

While the YSR features have a large bias asymmetry in the top panels, the main asymmetry in the

bottom panels seems to come from multiple Andreev reflections at small voltages. Is it possible to further analyze this regime? Does one expect a relation between the asymmetries in both cases?

- Fig. 4:

As the switching current displays some asymmetry as well, it would be useful to show a similar inset as for the retrapping current.

- In the theoretical model, both an asymmetric current-voltage characteristic and an asymmetric current-phase relation are considered. Is there an intuitive explanation as to which effect will more strongly affect the switching or retrapping current? Does something interesting happen in the presence of both?

Referee #3 (Remarks to the Author):

In this paper, the authors demonstrated a Diode-like effect in Josephson junctions with a single magnetic atom. The authors further claim that it is possible to tune their properties via single-atom manipulation. The work is interesting but one can wonder about what they actually learned from this atomic scale system compared with other heterostructures that show similar behavior. Importantly, how general is this controllable diode effect (magnitude and sign)? is it only applicable to atomic scale junctions?. At the moment, I can't see any significance of this work to guarantee the manuscript publish in Nature.

there are a few critical points that were not carefully accounted for in the paper, which are detailed below

1. The peak intensity and position of YSR states are governed by the scalar potential V and exchange coupling J , respectively. However, it is not clear from this experiment, which one is playing an important role. what controls the magnitude and sign of the diode effect in the system, V or J ? is there any relation between the magnitude and sign of the diode effect with J or V ?
2. what happens when you have a symmetric YSR ? is the author still observing a hysteresis behavior? is it on the electron side or the hole side of the spectra? the author should include this measurement in the manuscript.
3. The author mentions that this is a field-free diode-like effect due to the single magnetic atom. What happens when you apply an external magnetic field in the junction (below the critical field). this measurement should be included in the manuscript also.
4. Similar field-free diode behavior has been observed in a $\text{NbSe}_2/\text{Nb}_3\text{Br}_8/\text{NbSe}_2$ junction, However, the author claims that atomic-scale junctions differs qualitatively from observations of non-reciprocity in larger-scale junctions. The author did not give any explanation for why. The author should note that Nb_3Br_8 is not a ferromagnet, therefore, the junction is time-reversal symmetric similar to the system that the author has.
5. based on the author's previous work (PRL 115, 087001 (2015)), temperature also plays an important role. what is the role of temperature in this case?
6. the theoretical mode that the author used here is phenomenological and it does not capture YSR. but the experimental interpretation is related to YSR states. could the author comment on this?

Author Rebuttals to Initial Comments:

Color code:

Referees' comments (blue)

Our response (black)

Reference to changes (orange)

Referee #1 (Remarks to the Author):

Using scanning tunneling spectroscopy, the authors report observation of diode effect in Josephson junctions that contain single magnetic atoms. In the current biased spectroscopy measurements, they observe non-reciprocal behavior in both switching as well as retrapping currents, however, non-reciprocity is stronger in the retrapping current which is different compared to recent diode effects observed in other systems. While non-reciprocity in other systems has been ascribed to the breaking of inversion and time-reversal symmetry, authors claim that the observed non-reciprocity is due to sub-gap states resulting from the interaction between magnetic atoms with the substrate which drives asymmetric damping of the phase particle during retrap. Their claim is based on an extended RCSJ model that includes quasiparticle current via sub-gap states due to the magnetic impurity and they also exclude asymmetric current-phase relation which mainly affects the switching current. In view that the superconducting diode effect has recently gained interest due to its potential in realizing high-speed electronics without dissipation of the currents, this manuscript stimulates additional ways to realize the superconducting diode effect. Therefore, I recommend considering publication in Nature after the following concerns are clarified.

We thank the referee for carefully reading our manuscript and appreciating our results. We reply to his/her comments in the following.

1) What are the dark circles closeby the magnetic atoms in the inset image of Fig. 1a. Can authors rule out their role in the observed spectra?

The dark circles in the STM images arise from Ne inclusions below the surface. The sputtering process leads to implantation of Ne atoms, which coalesce and are then covered by Pb during the subsequent annealing process. These inclusions are well known for Pb (and Al) surfaces [Phys. Rev. Lett. 76, 2298 (1996), Phys. Rev. Lett. 114, 157001 (2015)]. They are inert and do not influence the bulk properties of Pb, but modify the electronic structure by inducing *local* quantum well states between the Ne “bubble” and the surface.

The quantum well states can be directly resolved in tunneling maps and our magnetic adatoms are always located at a sufficiently large distance from the Ne inclusions to exclude a substantial influence on the Josephson coupling. We also note that according to our measurements, the superconducting gap remains unaffected by the Ne inclusions.

This conclusion is supported quite explicitly by our extensive data. We investigated many Mn and Cr atoms at various distances from the Ne inclusions, stretching over several measurement times and using different tips. Within the statistical fluctuations, we did not observe any discernible influence of the distance on the asymmetries of the switching and retrapping currents.

To further exclude any influence of the Ne impurity, we include additional data for the Referee in Figure 1. Apart from Pb atoms showing a bistability on the Ne impurity, the switching and retrapping currents are identical within the statistical fluctuations. Importantly, there is no asymmetry in the retrapping current.

Figure 1 : a) STM image of a Pb(111) surface after deposition of Mn atoms and Pb adatoms. The buried Ne impurities can be seen as depressions of different sizes. b, c) Close-up views of two Pb atoms. The atom in (b) has been manipulated from the edge of the Ne impurity in (a) to the center of the Ne impurity. The Pb atom in (c) is isolated on the clean surface. d) Retrapping currents and e) switching currents as a function of the phase diffusion conductance as measured on the two Pb adatoms shown in (b) and (c) (red: Pb atom on Ne impurity; gray: Pb atom on clean surface). The retrapping currents for the Pb atom on the Ne impurity fall onto two lines, indicating a bistable adsorption configuration. Apart from that, the Pb atoms in the two locations behave similarly. Importantly, no asymmetries in the retrapping currents arise due to the Ne impurity.

2) All junctions show phase diffusion conductance which leads to finite dissipation. Are there ways to improve the quality of junctions to minimize/remove phase slip events?

As correctly pointed out by the referee, all our junctions show phase diffusion. The coexistence of phase slips and hysteresis is a known effect in small Josephson junctions as emphasized and explained in a classic paper by Kautz and Martinis [Phys. Rev. B 42, 9903 (1990)]. The phase slips are driven by finite temperature and noise, and can coexist with hysteretic behavior when the phase dynamics is subject to frequency-dependent damping due to the external electromagnetic circuit. Phase-diffusion processes can in principle be further suppressed by reducing temperature and noise as well as by modifying the external circuitry. While we cannot lower the temperature in our STM, we have made substantial efforts to suppress external noise sources.

We would like to emphasize that the existence of phase diffusion is not related to or required for the diode behavior occurring in our junctions:

- As for the influence of finite temperature and noise, we can exclude this based on our experimental data. We observe variations in the noise level over time, depending, e.g., on variations of the tip stability. This is reflected in small variations of the phase diffusion rate as well as the magnitude of the switching and retrapping currents (see Supplementary Figure 1a and Supplementary Figure 2 for additional data sets with effectively different noise levels in Pb adatom junctions). Importantly, however, these variations in noise never affect the sign of the asymmetry in the switching and retrapping currents, emphasizing that our qualitative conclusions are robust and independent of the phase-diffusion rate (see Supplementary Note 4 for data taken with different tips and effectively different noise levels).
- We also note that we always compare our measurements on magnetic adatoms with reference measurements on non-magnetic Pb adatoms, which also exhibit phase diffusion.
- As for the influence of frequency-dependent damping, this follows from our theoretical modeling. The theoretical simulations shown in the manuscript account for frequency-dependent damping and hence reproduce a finite phase-diffusion rate. However, we can readily simulate junctions without frequency dependence of the damping. These simulations lead to junction dynamics *without* phase diffusion. We find that the asymmetry in the retrapping current persists, while the slight asymmetry in the switching-current histograms is actually reduced. For the benefit of the referee, Figure 2 shows corresponding representative simulation data based on a fully microscopic model of a Josephson junction with YSR state.

[This has been redacted]

How do current biased spectra look on bare Pb substrate?

It is much more difficult to establish stable junctions on the bare Pb substrate. For junctions on the bare substrate, mechanical noise leads to effective variations in the junction conductance, making reliable measurements at reproducible conductance values almost impossible. The junctions including a Pb adatom are much more stable. We speculate that at these high junction conductances, the forces between the tip and a flat substrate induce instabilities of the tip apex. A single adatom protruding from the surface reduces the long-range forces and thereby stabilizes the junction.

We note that it is actually preferable to create junctions with a Pb adatom. Apart from the chemical nature of the adatom, these junctions are structurally equivalent to the junctions with a magnetic

atom. Thus, identical conductance values correspond to similar tip-adatom distances, making a direct comparison much more meaningful.

Does the finite size of the tip-superconductor induce additional contribution to the Gpd ? The last one can be checked by measuring the width of the Josephson peak and seeing if it is broadened by the temperature alone.

The finite size of the tip can in principle affect the junction dynamics in two ways:

- First, it may affect the superconducting gap in the tip, reducing it below the bulk gap of Pb. For our junctions, we can exclude finite-size effects on the superconducting gap of the tip. Differential conductance spectra on the bare Pb substrate show a clean gap with zero conductance. The gap in the differential conductance spectra as delimited by the BCS coherence peaks extends to voltages of $\pm 2\Delta_{\text{Pb}}$ to excellent accuracy. As this is expected to be equal to the sum of the tip and substrate gaps, this measurement confirms directly that the superconducting gap of the tip is equal to the bulk gap Δ_{Pb} of Pb.
- Second, the physical size of the tip governs the capacitance of the junction, which is an important parameter in determining whether the junction dynamics is classical or quantum. However, our junctions are safely in the classical regime, given that the measurements are taken at a temperature of 1.3K. Capacitances of STM junctions are expected to be of order 1-50fF, depending on details of the tip geometry [see, e.g., Ast et al., Nature Comm. **7**, 13009 (2016)]. With our parameters, this implies plasma frequencies ω_p in the range of 10-100GHz. At the same time, quantum tunneling becomes relevant below a crossover temperature $T_x \simeq \frac{\hbar\omega_p}{2\pi k_B}$, corresponding to 0.1-1K. Moreover, this estimate is an upper limit since the junction is subject to instrumental noise in addition to thermal fluctuations. This is consistent with the width of the Josephson peak, which is of the order of, but exceeding the expected thermal broadening of $\left(\frac{2e^2}{\hbar}\right) k_B T Z$ [see, e.g., Grabert & Ingold, arXiv:9811194]. (Here, Z is the junction impedance at high frequencies, which is typically significantly lower than and essentially independent of the dc resistance R of the junction.) We interpret the additional broadening as due to additional instrumental noise.
For the benefit of the referee, we include corresponding data in Fig. 3. Consistent with theoretical expectations, the width of the Josephson peak is hardly affected by the dc junction resistance or the adatom species. Detailed quantitative comparisons are complicated by the fact that the high-frequency impedance Z is not known precisely.

[This has been redacted]

[This has been redacted]

We now emphasize on page 8 of the manuscript that our junctions are well described by a classical model.

3) The authors mention about biasing junctions with smaller rates during the spectroscopy. Can authors comment on the response of the junctions with faster switching and if/how the finite G_{pd} can limit the switching speed?

Both the switching and the retrapping current are well-known to depend on the ramp rate. However, these dependencies are logarithmic and thus rather weak. We have varied the ramp rate within some range and observed changes in the retrapping and switching currents which are consistent with this weak dependence and follow the expected trends. While the (average) switching current tends to decrease with increasing ramp rate, the (average) retrapping current increases.

We note that we used similar ramp rates (100 nA/s to 320 nA/s) for the measurements throughout the manuscript. For the individual data sets using the same tip on the magnetic atoms and the reference Pb adatom, we used the same ramp rate to ensure a useful comparison between the measurements.

The influence of the nonzero phase diffusion on the switching rate is actually a rather involved (and to the best of our knowledge: unsolved) theoretical problem. Some aspects are discussed in the above-mentioned paper by Kautz and Martinis. These authors point out that frequency-dependent damping (underlying the non-zero phase diffusion) introduces a phase-space structure admitting direct transitions between attractors corresponding to neighboring minima of the tilted washboard potential. This allows phase slips to coexist with switching events into the resistive state, but the complicated phase-space structure along with the frequency dependence inhibits theoretical progress.

We added a comment on the dependence on the ramp rate in the methods section.

4) Authors quoted in the manuscript (line 195) and supplementary (page 7) that simulations for Pb do not show asymmetry in the switching/retrapping currents. However, Fig 4 clearly shows a large asymmetry in the switching. Can the authors clarify the discrepancy in the simulation?

The switching and retrapping events are induced by fluctuations. As a result, the switching and retrapping currents are statistical quantities that depend on the particular current ramp. For this reason, an individual current ramp such as the one shown in Fig. 4 generically exhibits asymmetry in both the retrapping current and the switching current. The presence or absence of nonreciprocity is only established after many current ramps and mapping out the entire histograms of the switching and retrapping currents. As shown in the supplement for both the experimental data (Supplementary Fig. 1) and the simulations (Supplementary Fig. 7), the histograms are entirely consistent with our claims. Simulations of the Pb junctions yield histograms of the switching and retrapping currents, which are symmetric. In contrast, the histograms are asymmetric in the simulations for the magnetic adatoms, with the retrapping currents being much more asymmetric than the switching currents.

We now explicitly point out the importance of the statistical nature of the switching current in the main text (page 11).

In addition, authors can consider the following optional comments.

1) Owing to the large asymmetry in the observed currents, authors can consider quantifying the diode efficiency.

The diode efficiency depends on junction resistance as can be seen from the different slopes of switching and retrapping currents in Fig. 2. The asymmetry of the junction thus increases with increasing junction conductance. Hence, the diode efficiency is not a well-defined number that is comparable to other types of junctions. We would therefore prefer to omit a specific value.

2) What happens to the junction characteristics when the time-reversal symmetry is broken (if they have access to in-plane magnetic fields)?

Unfortunately, Pb has a very small critical field (80 mT). Thus, the superconducting gap will close long before the magnetic impurity is significantly affected. Since the substrate is a bulk crystal, this remains true for in-plane magnetic fields.

Based on our theoretical understanding, we do not expect a significant change when polarizing the adatom spin (thereby breaking time reversal symmetry). In fact, the same asymmetry behavior comes out theoretically within a fully microscopic model for a polarized adatom spin (see also our response to referee #3 below). In principle, time reversal breaking in conjunction with spin-orbit coupling can induce an asymmetric current-phase relationship in addition to asymmetric damping. This would give rise to an additional nonreciprocity in the switching current, which essentially adds to the asymmetry in the retrapping current.

It is an important point of our paper that the dominant diode behavior in the retrapping current is independent of whether time reversal symmetry is broken or not. It is for this reason that it can be observed without application of a magnetic field. Instead, we show that dominant diode behavior in the retrapping current is rooted in broken particle-hole symmetry due to potential scattering by the magnetic impurity. This potential is reflected in asymmetric weights of the particle and hole excitation of the Yu-Shiba-Rusinov states, inducing asymmetric damping.

3) Did authors try similar spectroscopy off the atoms on bare Pb in the vicinity of magnetic atoms and especially close to the dark impurities?

As explained above, junction instability does not allow us to measure Josephson junctions, in which the tip is off the adatoms and faces the bare surface. This is particularly true since meaningful measurements (switching and retrapping current histograms) require stability over hundreds of current sweeps.

However, we created Pb adatom junctions in close vicinity to magnetic adatoms for direct comparison. To do so, we manipulated a Pb atom across the surface with the STM tip. In this way, the Pb atom can be pushed into lattice sites at different distances from the magnetic adatom (though always at equivalent sites of the substrate's atomic lattice). We observed that already at the closest possible distance of two lattice sites (a closer distance results in ill-defined clusters), measurements on the Pb atoms give essentially the same behavior as on an isolated Pb atom.

As mentioned above, the dark impurities are Ne inclusions, which neither affect the superconducting properties nor the Josephson junctions established on Pb atoms. We do not have data on magnetic atoms on the Ne impurities, since unlike the Pb adatoms, Mn and Cr cannot be manipulated by the STM tip (probably due to their smaller size and larger binding energies). However, based on the results for Pb junctions, we would not expect a significant influence either.

4) There is an error at the beginning of line 265.

For better readability we rewrite: **Identical tip and identical tip location**.

Referee #2 (Remarks to the Author):

The authors use an STM to study Josephson junctions through an individual magnetic adatom. They observe an asymmetry of the retrapping currents that depends on the type of adatom. The switching currents display a much weaker asymmetry. The results are presented in the context of so-called Josephson diodes that have received a lot of attention recently. While the asymmetry in these Josephson diodes arises due to an interplay of broken inversion symmetry and broken time-reversal symmetry, the authors attribute their observations to a different mechanism, namely an interplay between broken inversion symmetry and broken particle-hole symmetry. While the energy spectrum of the junction is particle-hole symmetric, the wavefunctions of the YSR states forming in the junctions are not, leading to different intensities in tunneling spectroscopy at positive and negative bias - an effect that is well known, but to my knowledge has not been linked to asymmetries in switching/retrapping currents before. The authors back up their claim using numerical simulations based on the RCSJ-model and taking an asymmetric current-voltage characteristic as an input.

The results are interesting and very timely. To recommend them for publication in Nature, the main message would have to be clarified though.

We thank the referee for carefully reading our manuscript and his/her appreciation of our results. We tried to clarify the message of the manuscript by mainly modifying the introduction (see also the next point) and reply to his/her detailed comments in the following.

While the paper is well written, I find the abstract and introduction somewhat misleading: they suggest that what the authors achieved is a miniaturization of the Josephson diode as discussed in earlier works - before then evoking a completely different mechanism. The work should be set in a broader context and mention other possible reasons for asymmetries (such as self-field effects). The conclusion as well stresses the miniaturization aspect, even though it doesn't seem to me that the difference in mechanism is due to the size of the junction. It would be useful to distinguish (or relate more clearly) the two aspects.

The main point of our paper is indeed that we discovered a new mechanism leading to diode behavior in Josephson junctions. We therefore agree with the referee that we can strengthen this aspect **and have followed this advice by changing the abstract and introduction accordingly**.

However, we also find it intriguing and important that this new mechanism operates already when including a single magnetic atom into a Josephson junction. One can easily exchange the atomic species and thereby manipulate strength as well as direction of the diode effect. We further envision that few-atom structures with interesting magnetic textures could provide additional tuning parameters.

Some more specific comments/questions:

- Fig. 1:

For some values of G_{PD} (above 60 μS), there are two different data points. Why?

Figure 2 shows different data points for the same G_{PD} , because we have repeated the measurements on different junctions. As there are some statistical fluctuations due to small tip changes and changes in the noise level over time, the results may differ slightly between the measurement series. We have collected these data in Fig. 2 to expose these slight variations.

We now explicitly mention the different sets in the figure caption.

Furthermore, for the clarity of the figure, it might be better to use a larger contrast between dark and bright colors.

We have changed the contrast for better visibility.

- Fig. 3:

While the YSR features have a large bias asymmetry in the top panels, the main asymmetry in the bottom panels seems to come from multiple Andreev reflections at small voltages. Is it possible to further analyze this regime? Does one expect a relation between the asymmetries in both cases?

There is indeed a relation between the asymmetry at far and close tip-sample distance:

At small junction transparency, the current is carried by single-electron tunneling. Given a particular YSR state of energy ε_0 , single-electron tunneling is restricted to bias voltages $eV > \Delta + \varepsilon_0$, where Δ is the superconducting gap of the tip. Consequently, the asymmetry is restricted to the vicinity of the threshold voltage $eV = \Delta + \varepsilon_0$ and arises from the asymmetry in the electron and hole wave functions of the YSR state. At larger junction transparencies, the current is increasingly carried by multiple Andreev processes. These contribute already at lower voltages $eV > 2\Delta/n$ and $eV > (\Delta + \varepsilon_0)/n$, where n indicated the number of Andreev reflections required for the multiple Andreev reflection. Here, the first condition is for a process which leaves a quasiparticle in the above-gap quasiparticle continuum, while the second condition is for processes which occupy the YSR excitation in the final state. Due to the asymmetric electron and hole wave functions of the YSR state, the latter also lead to an asymmetric quasiparticle current.

Thus, one expects that multiple Andreev processes lead to asymmetries in $\frac{dI}{dV}$ over a much more extended voltage range, i.e., already at bias voltages below Δ_{tip}/e . Concurrent with the emergence of symmetric multiple Andreev processes, the asymmetry is correspondingly reduced. We are not aware of a corresponding theoretical (or experimental) analysis of the precise tunneling rates and associated intensities, although we agree with the referee that this might be an interesting problem for future research. The asymmetries at high junction transparency can in principle be extracted by analyzing the tunneling process. However, the processes are no longer well separated at high junction conductance, making their experimental identification less straight forward than in the weak-tunneling limit. It is precisely for this reason that we included tunneling spectra for junctions at weak tunneling, although the regime of stronger tunneling is relevant for the Josephson measurements.

- Fig. 4:

As the switching current displays some asymmetry as well, it would be useful to show a similar inset as for the retrapping current.

As the figure in the main text would be overcrowded by an additional inset, we include a figure showing the absolute values of switching and retrapping currents in the Supplementary Material.

- In the theoretical model, both an asymmetric current-voltage characteristic and an asymmetric current-phase relation are considered. Is there an intuitive explanation as to which effect will more strongly affect the switching or retrapping current? Does something interesting happen in the presence of both?

The intuition behind this observation is the following. In the absence of fluctuations, the junction switches at the critical current, i.e., at the current bias at which the tilt of the washboard potential eliminates the minima. In this limit, an asymmetry in the switching current requires an asymmetric washboard potential, or equivalently, an asymmetric current-phase relation. Fluctuations will then reduce the switching current below the critical current, but in the limit of weak damping (pronounced hysteresis), the asymmetry of the switching current is largely inherited from the asymmetry in the critical current.

The retrapping current, on the other hand, is the result of quite different physics. In the absence of fluctuations, the junction retraps, once the energy gain due to the bias current (or due to the tilt in the language of the washboard potential) becomes smaller than the frictional energy loss during the motion. The energy gain depends on the tilt, but not on the shape (or the asymmetry) of the washboard potential. Thus, an asymmetry in the retrapping current can only arise from asymmetries in the frictional energy loss, which is associated with the quasiparticle current at the microscopic level. Fluctuations tend to increase the retrapping current, but the asymmetry is essentially inherited from the asymmetry in the retrapping currents of the junction in the absence of fluctuations.

When both effects are present, they essentially simply add.

We have now added a new section at the beginning of Supplementary Note 5, which discusses the role of the exchange and potential scattering as well as the symmetries in more detail.

Referee #3 (Remarks to the Author):

In this paper, the authors demonstrated a Diode-like effect in Josephson junctions with a single magnetic atom. The authors further claim that it is possible to tune their properties via single-atom manipulation. The work is interesting but one can wonder about what they actually learned from this atomic scale system compared with other heterostructures that show similar behavior. Importantly, how general is this controllable diode effect (magnitude and sign)? is it only applicable to atomic scale junctions?. At the moment, I can't see any significance of this work to guarantee the manuscript publish in Nature.

We thank the referee for carefully reading the manuscript and the interest in our work. Apparently, the significance of our work – in particular the discovery of a new mechanism of the diode effect –

went unnoticed. This may be partly attributed to our previous introduction, which was apparently emphasizing the single-atom nature of our junction over the new mechanism (see also Referee #2).

In brief, we experimentally discovered and theoretically explained diode behavior in the retrapping current in a Josephson junction. In contrast to previous works on heterostructures, we find the largest asymmetry in the retrapping rather than the switching current. We attribute this behavior to broken particle-hole symmetry in the junction. This is a new effect and a widely applicable mechanism.

In addition to the discovery of a new mechanism, we believe that it is another beauty of our work that diode behavior can already be induced by a single magnetic adatom in a Josephson junction, which is otherwise made of an elemental superconductor.

Furthermore, a single-atom junction offers considerable flexibility compared to heterostructures. One easily controls the junction resistance or changes the adatom species (thereby changing strength and direction of the diode effect). One can also imagine assembling few-atom structures with the additional freedom of creating magnetic textures.

We are convinced that the discovery and thorough explanation of the effect is a significant advance and of interest for a broad audience.

We have rewritten the abstract and introduction to further highlight that we discover a new mechanism for non-reciprocity and to emphasize the differences with previous works.

there are a few critical points that were not carefully accounted for in the paper, which are detailed below

1. The peak intensity and position of YSR states are governed by the scalar potential V and exchange coupling J , respectively. However, it is not clear from this experiment, which one is playing an important role. what controls the magnitude and sign of the diode effect in the system, V or J ? is there any relation between the magnitude and sign of the diode effect with J or V ?

The referee is correct that the exchange and potential scattering amplitudes are important parameters. As we explain in the manuscript, the diode effect in the retrapping current can be traced to asymmetries in the quasiparticle current flowing via the YSR resonances. YSR resonances are asymmetric only when potential scattering is nonzero. Consequently, the diode effect in the retrapping current will also appear only when potential scattering is nonzero. Moreover, the sign of the asymmetry in the retrapping current is associated with the sign of the potential scattering amplitude V .¹

In the course of follow-up theoretical work, we have simulated current-biased YSR Josephson junctions, using a microscopic model incorporating both exchange and potential scattering (without frequency-dependent damping). We include the resulting histograms for the retrapping and switching currents for the benefit of the referee:

¹ Here, we use the referee's notation V for the potential-scattering amplitude. This should not be confused with the notation of the manuscript, where V represents the voltage. (Here, we use a lower-case v for voltage.)

[This has been redacted]

The magnitude of the diode effect depends in an intricate manner on J and V . On the one hand, the asymmetry grows with $|V|$. On the other hand, the asymmetry is also controlled by the energy of the YSR resonance, which varies with J and V as well. Typically, one would expect a stronger diode effect for smaller YSR energies, which leads to asymmetries in the quasiparticle current at lower voltages. We note that these theoretical expectations are based on a simplified model of a spin-1/2 impurity. Real magnetic impurities such as Cr and Mn presumably have larger spins and couple to multiple conduction electron channels. Each of these channels is characterized by its own exchange and potential scattering strength, resulting in the multiple YSR states, which we observe. A full theoretical description of these aspects would be rather complicated, but the basic qualitative physics should already be well described by the spin-1/2 model.

We have now added a new section at the beginning of Supplementary Note 5, which discusses the role of the exchange and potential scattering as well as the symmetries in more detail.

2. what happens when you have a symmetric YSR ? is the author still observing a hysteresis behavior? is it on the electron side or the hole side of the spectra? the author should include this measurement in the manuscript.

As we already showed in response to the previous question, a symmetric YSR resonance implies that there is no diode effect. At the same time, our simulations confirm that there is hysteresis, as of course expected for any weakly-damped Josephson junction.

Measurements for a symmetric YSR states are not available as the potential scattering of a magnetic adatom on a particular substrate cannot be tuned experimentally.

3. The author mentions that this is a field-free diode-like effect due to the single magnetic atom. What happens when you apply an external magnetic field in the junction (below the critical field). this measurement should be included in the manuscript also.

The experiment is indeed done in the absence of a magnetic field. We can also rule out that the adatom spin is effectively polarized, for instance due to strong single-ion anisotropy. If this polarization was the origin of the asymmetry, the sign of the non-reciprocity should depend on the orientation of the adatom spin. Consequently, we should occasionally observe non-reciprocities of opposite sign in different measurement times or even changes of the non-reciprocity during a

particular measurement time. However, we observe that a particular adatom species always exhibits non-reciprocities of a *fixed* sign.

Experimentally, the substrate is a bulk Pb crystal with a minute critical field. For this reason, a magnetic field would close the superconducting gap long before significantly affecting the adatom spin. We thus do not believe that measurements in a magnetic field are very revealing.

We also note that theoretically, we do not expect our mechanism for a non-reciprocal retrapping current to be sensitive to an applied Zeeman field. In fact, the simulations included above are done for a microscopic model with a *polarized* magnetic impurity, but exhibit the same qualitative behavior as the phenomenological simulations in the manuscript where there is no polarization of the adatom spin.

4. Similar field-free diode behavior has been observed in a NbSe₂/Nb₃Br₈/NbSe₂ junction, However, the author claims that atomic-scale junctions differs qualitatively from observations of non-reciprocity in larger-scale junctions. The author did not give any explanation for why. The author should note that Nb₃Br₈ is not a ferromagnet, therefore, the junction is time-reversal symmetric similar to the system that the author has.

As we write in the manuscript, our observations differ qualitatively from larger-scale NbSe₂/Nb₃Br₈/NbSe₂ junctions. While in the latter, the dominant nonreciprocity is in the switching current, we observe a dominant nonreciprocity in the retrapping current. As we show in the manuscript, an asymmetric retrapping current is naturally explained in terms of an asymmetry in the dissipative quasiparticle current flowing in parallel to the supercurrent. We speculate that the NbSe₂/Nb₃Br₈/NbSe₂ junctions have a much weaker or no asymmetry of the quasiparticle current and the observed nonreciprocity is due to a different effect.

We now emphasize the difference with previous works in the introduction.

5. based on the author's previous work (PRL 115, 087001 (2015)), temperature also plays an important role. what is the role of temperature in this case?

In our previous work, we analyzed the transport processes through YSR state. Essentially, excitation of the YSR states by single-electron tunneling needs to be followed by a relaxation process. Otherwise, there would be no current flow through the subgap states. At far tip-sample distance, the tunneling rate is small and the relaxation processes are sufficiently fast such that each tunneling electron finds the YSR excitation in its ground state. The asymmetry of the dI/dV spectrum is then controlled by the ratio of the electron- and hole wavefunctions of the YSR state. Upon tip approach, the tunneling rate increases and eventually becomes larger than the relaxation rates. At this point, resonant Andreev reflections start to carry a significant part of the current. The differential conductance given by these processes is of opposite bias asymmetry in the dI/dV spectra (for superconducting tips!), because both electron- and hole-components of the wavefunction are involved in the transport. Temperature enters through the temperature dependence of the relaxation rates.

In the present experiments, we are however in a regime of junction conductances, where multiple Andreev processes already contribute significantly to the differential conductance. As described in response to Referee #2, these also lead to an asymmetry in the low-voltage regime due to excitation of the YSR states at bias voltages $eV > (\Delta + \varepsilon_0)/n$, where n denotes the number of Andreev

reflections. We expect that this regime is less affected by temperature as the excitation rates are much smaller due to the larger number of involved Andreev reflections.

While the differential conductance is thus less sensitive to temperature, temperature enters the junction dynamics via thermal fluctuations. It is thermal fluctuations which cause phase diffusion and affect the switching and retrapping currents. A larger temperature results in a decrease of the switching current, reflecting a larger probability for escaping from the minima of the washboard potential. It also results in an increase of the retrapping current.

While we cannot measure at lower temperatures, larger temperatures are in principle possible. However, junction stability would limit the accessible temperature range, so that we do not expect to learn much from these experiments.

6. the theoretical mode that the author used here is phenomenological and it does not capture YSR. but the experimental interpretation is related to YSR states. could the author comment on this?

The theory does not describe the YSR states microscopically. However, the physics of the subgap conductance in the presence of magnetic adatoms has been widely studied and the observed subgap conductance is clearly associated with the fact that the magnetic adatom induces YSR states within the superconducting gap. Our phenomenological approach allows us to extract the relevant quasiparticle current directly from experiment and to simulate its effects on the junction dynamics. When combined with the well-developed knowledge on the subgap conductance due to YSR resonances, this gives a rather complete picture of the origin of the nonreciprocal behavior in our junctions.

As indicated by the simulation data included above in this response, it is certainly possible to develop a more microscopic theory. This microscopic theory completely confirms all conclusions of the present manuscript. However, while such a theory gives additional insight, it remains quite far from the experimental system: (a) It only accounts for single-electron tunneling, while multiple Andreev processes are clearly relevant in experiment. (b) It treats the magnetic impurity as a classical spin inducing a single pair of YSR resonances, while the magnetic adatoms induce multiple YSR states in experiment. Developing a complete microscopic theory which can be directly compared to experiment is a rather difficult endeavor. We believe that our phenomenological modelling is thus very much appropriate for interpreting the experimental data.

We realize, however, that we may not have provided sufficient background knowledge to make the relation with YSR states and their asymmetries transparent. We have now added a new section at the beginning of Supplementary Note 5, which discusses these issues in more detail.

Reviewer Reports on the First Revision:

Referees' comments:

Referee #1 (Remarks to the Author):

The authors provided reasonable explanations for all of my concerns and clarified all the doubts in their reply and the revised manuscript. In particular, in view of dissipationless electronics, their new simulation in Figure 2 of the reply regains confidence in building low power circuits, and in my view, this is an important result which should be part of the paper/supplement for better outlook. It remains to be seen in case the research following this paper ensures that the spin states of the magnetic atom/s in the junction are preserved and that the junction exhibit proposed V-I characteristics. I thus, with this optional modification, recommend publishing the paper in Nature without any further delay.

Referee #2 (Remarks to the Editor)

Referee #2 still thinks that it would be important to place the results in a broader context. Namely other mechanisms/setup for critical current asymmetries (self-field effects, asymmetric SQUIDs, ...) have been known even before Josephson diodes became popular in recent years and should be cited.

Referee #3 (Remarks to the Author):

The authors addressed my questions properly. I would like to recommend it publish in Nature, although I still concern a little about the significance of the work.

Author Rebuttals to First Revision:

Referee #1 (Remarks to the Author):

The authors provided reasonable explanations for all of my concerns and clarified all the doubts in their reply and the revised manuscript. In particular, in view of dissipationless electronics, their new simulation in Figure 2 of the reply regains confidence in building low power circuits, and in my view, this is an important result which should be part of the paper/supplement for better outlook. It remains to be seen in case the research following this paper ensures that the spin states of the magnetic atom/s in the junction are preserved and that the junction exhibit proposed V-I characteristics. I thus, with this optional modification, recommend publishing the paper in Nature without any further delay.

We thank the referee for recommending our work to be published in Nature.

The theoretical work, which we included with the response would require us to add several pages to the supplementary material with the details of the calculation to make it reproducible, in addition to a discussion of the results (most of which would also have to be included with the supplemental material). This makes it more adequate to publish the calculation as part of a more extensive theoretical study.

Referee #2 (Remarks to the Editor)

Referee #2 still thinks that it would be important to place the results in a broader context. Namely other mechanisms/setups for critical current asymmetries (self-field effects, asymmetric SQUIDs, ...) have been known even before Josephson diodes became popular in recent years and should be cited.

As pointed out by the Referee, several superconducting devices such as Josephson amplifiers and ratchets made use of asymmetric current-phase relationships in *two- and more* junction devices (as opposed to the *single*-junction devices which are currently under study). We agree that a review of these phenomena and implementations would be a valuable contribution by providing context to the recent developments in the field.

While a more thorough review is evidently beyond the scope of the present paper, we now mention two works to alert readers of this broader context. We have added: "While two or more Josephson junctions combined into SQUIDS have long been proposed as amplifiers and rectifiers [13,14], experiments on single Josephson junctions have only recently observed non-reciprocal behavior.

Referee #3 (Remarks to the Author):

The authors addressed my questions properly. I would like to recommend it publish in Nature, although I still concern a little about the significance of the work

We thank the referee for recommending our work to be published in Nature.